# RNA polymerase II pausing is essential during spermatogenesis for appropriate gene expression and completion of meiosis

Emily G. Kaye [1], Kavyashree Basavaraju [2], Geoffrey M. Nelson [1], Helena D. Zomer[2], Debarun Roy[2], Irene Infancy Joseph[2], Reza Rajabi-Toustani[2], Huanyu Qiao [2], Karen Adelman [1] ✉ & Prabhakara P. Reddi [2] ✉

Male germ cell development requires precise regulation of gene activity in a cell-type and stage-specific manner, with perturbations in gene expression during spermatogenesis associated with infertility. Here, we use steady-state, nascent and single-cell RNA sequencing strategies to comprehensively characterize gene expression across male germ cell populations, to dissect the mechanisms of gene control and provide new insights towards therapy. We discover a requirement for pausing of RNA Polymerase II (Pol II) at the earliest stages of sperm differentiation to establish the landscape of gene activity across development. Accordingly, genetic knockout of the Pol II pause-inducing factor NELF in immature germ cells blocks differentiation to spermatids. Further, we uncover unanticipated roles for Pol II pausing in the regulation of meiosis during spermatogenesis, with the presence of paused Pol II associated with double-strand break (DSB) formation, and disruption of meiotic gene expression and DSB repair in germ cells lacking NELF.

Mammalian spermatogenesis is a highly conserved and carefully orchestrated cell differentiation process. Understanding male germ cell development is of paramount importance since defects in spermatogenesis typically result in a failure to produce spermatozoa and infertility. Male infertility is a major reproductive health issue affecting at least 30 million men globally with limited treatment options available[1]. Although the major cell types involved in spermatogenesis and the key transitions between these stages have been described, the asynchronous nature of spermatogenesis within the seminiferous tubules and resulting cellular heterogeneity has confounded efforts to understand the mechanisms governing the changes in gene expression, chromatin structure and cell morphology that accompany sperm development[2].

Male germ cell development starts with the spermatogonial stem cells and culminates in the formation of spermatozoa. This process, which involves three major developmental stages, is conserved between mice and humans[3]. Spermatogonia (SG) divide by mitosis, differentiate, and commit to meiosis. Spermatocytes

(SC) go through two rounds of meiotic division and give rise to haploid round spermatids (RS). The RS then enter terminal differentiation, acquiring unique structures such as the acrosome and flagellum to become elongated spermatozoa. Throughout the process of germ cell development, somatic support cells such as Sertoli and Leydig cells provide signaling molecules to maintain the stem cell niche[4].

Within the seminiferous epithelium, germ cells advance through spermatogenesis as cohorts transitioning through a series of developmental stages. This suggests a requirement for precise spatio-temporal gene regulation within each cell type of a cohort[5]. Consistent with this, perturbation of transcription and RNA binding factors such as TDP-43 cause severe defects in sperm maturation[6,7]. TDP-43 is a protein with multiple roles in RNA metabolism and transcription, and has been suggested to regulate the promoter-proximal pausing of Pol II[6–8]. However, the mechanisms underlying the defects in male germ cells lacking TDP-43 or other regulators of gene expression remain to be fully explored.

[1]Department of Biological Chemistry and Molecular Pharmacology, Harvard Medical School, Boston, MA 02115, USA. [2]Department of Comparative Biosciences, University of Illinois Urbana-Champaign, Urbana, IL 61802, USA. ✉e-mail: karen_adelman@hms.harvard.edu; preddi@illinois.edu

Here, we probed gene activity in purified SG, SC, and RS from murine testis. Through integration of steady-state RNA-seq with nascent RNA analysis using Precision Run-On Sequencing (PRO-seq), we reveal widespread changes in gene expression during spermatogenesis and demonstrate that this regulation occurs predominantly at the level of transcription. We find that the controlled pausing of Pol II in early transcription elongation plays a critical role in the proper expression of genes during male germ cell development. Specifically, we discover that the selective establishment of paused Pol II at promoters in SG is essential for appropriate gene activation in SC. Using immunohistochemistry and single cell RNA-sequencing (scRNA-seq) in testis lacking the Pol II pause-inducing complex NELF we demonstrate that loss of NELF in SG prevents appropriate differentiation and meiotic progression of SC, and dramatically reduces the number of RS produced. Further, we identify an intriguing connection between Pol II pausing and the location of double strand breaks created during meiosis. Together, our results reveal an essential role for NELF-mediated Pol II pausing in spermatogenesis and shed new light on the mechanisms underlying the requirement for TDP-43 in this process. Further, these data represent a comprehensive analysis of gene control during spermatogenesis, opening new avenues for understanding idiopathic male infertility and for the development of new treatments.

## Results

### Widespread changes in gene expression during spermatogenesis

To define the mechanisms underlying gene regulation during spermatogenesis we purified populations of SG, SC and RS derived from wild type C57BL/6 J mice. The SG were isolated from 6 to 8 day old mouse testis, while SC and RS cells were isolated from three-month-old mice using the STA-PUT method[7,9]. To validate our cell populations and characterize steady-state RNA levels, we performed total RNA sequencing (RNA-seq). Cells were spiked prior to RNA extraction to allow for accurate normalization of reads across conditions. Our normalization strategy further included a correction for the haploid status of RS cells as compared to diploid SG and SC stages (see Methods). Although SCs have 4C DNA content following replication in meiosis, we could not find evidence that all genome copies were competent for transcription at this stage, so no additional corrections were made. Inspection of individual loci confirmed the appropriate expression profiles of previously described marker genes for each cell type, validating our purification and normalization procedure. For example, *Crabp1* RNA expression is highest in SG (Fig. 1a), consistent with a role for this gene in the retinoic acid signaling that prepares SG to differentiate and enter meiosis[10,11]. Likewise, *Spo11*, which is essential in SC for the initiation of meiosis and double-strand break formation[12], exhibits the highest RNA expression in SC (Fig. 1b). The testis-specific kinase *Tssk3*, which is involved in spermiogenesis[13] and is a marker of RS, is expressed most strongly in this cell type (Fig. 1c).

Analysis of differentially expressed genes between SG and SC identified 9600 upregulated and 3764 downregulated genes (Fig. 1d), indicative of broad alterations in cell state during this transition. The transition from SC to RS also involved widespread alterations in the transcriptome; however, in this situation, the majority ($n = 12,269$) of differentially expressed genes are decreased in RS with far fewer genes induced ($n = 1135$) (Fig. 1e). This general downregulation of gene activity in RS is consistent with the silencing of transcription in preparation for the condensation of chromatin in late spermatids[14]. Notably, although previous work has reported considerable gene expression changes during spermatogenesis[11,15–17], our spike-in normalization allows us to detect more differential expression than has been previously observed, providing a comprehensive profile of gene activity in each cell type.

### Distinct clusters of expression across germ cell development

We next sought to identify groups of genes with similar expression profiles across spermatogenesis. We calculated the relative RNA-seq levels across cell types for each differentially expressed gene ($n = 17,078$ genes) and used these values to perform clustering analyses. This yielded six clusters with distinct expression dynamics (Fig. 1f). To characterize the genes in each cluster, we performed gene ontology (GO) term analysis. We found that genes in cluster 1 ($n = 3226$), which are highly expressed in SG but rapidly repressed during differentiation, are enriched in general housekeeping functions such as cell adhesion, morphogenesis, and cell growth (Fig. 1g). Cluster 2 genes ($n = 2645$), which are repressed more slowly than cluster 1 during the transition from SG to SC, represent additional housekeeping GO categories, including metabolic processes (Supplementary Fig. 1a). Cluster 3 ($n = 3601$), which becomes activated during differentiation of SG to SC, is enriched in genes governing mRNA processing and post-transcriptional regulation of gene expression (Supplementary Fig. 1b). This finding is consistent with reports that RNA processing and 3' end formation are altered in SC and RS[18–20]. Cluster 4 genes ($n = 4024$), which are selectively induced in SC, include GO terms important for meiosis, genome integrity, and sperm development such as the piRNA pathway and cilium assembly (Fig. 1h). Similarly, cilium organization is the top GO term represented in Cluster 5 ($n = 1596$) (Supplementary Fig. 1c), in agreement with SC cells beginning to establish the machinery required for sperm motility. Finally, cluster 6 genes ($n = 1986$) that have very low expression in SG but are highly activated in RS are enriched in the GO terms spermatid development and the acrosome reaction (Fig. 1i). Thus, these clusters reflect known germ cell biology, and expand upon this knowledge to encompass thousands of genes not previously appreciated to be regulated during spermatogenesis.

Although most clusters include predominantly protein-coding genes, lncRNAs are also represented among the differentially expressed genes. Intriguingly, cluster 6 genes are almost 40% lncRNA (Fig. 1j; $n = 762$ lncRNAs). This finding is consistent with the observed makeup of RNAs packaged within human sperm, which are enriched in lncRNAs[15,21,22]. Prior work has demonstrated that the cluster 6 gene *Acrv1* is coactivated in RS with the annotated lncRNA 1700027I24Rik[23,24]. This lncRNA is located upstream and antisense from the *Acrv1* promoter, suggesting that RS gene regulation might involve coordinated expression of mRNAs and proximal lncRNAs. Indeed, investigation of lncRNAs transcribed from within 1 kb of a protein coding gene revealed 31 mRNA-lncRNA pairs that are coordinately upregulated cluster 6 transcripts.

We also noted that genes expressed later in spermatogenesis tended to be shorter than those expressed earlier, with cluster 6 genes having significantly smaller distances from transcription start sites (TSS) to transcript end sites (TES) than genes in other clusters (Fig. 1k). This result held true when considering only mRNAs in each cluster (Supplemental Fig. 1d), confirming that the short length of many lncRNAs was not biasing this measurement. We speculate that genes expressed late in spermatogenesis may be preferentially short, to enable rapid RNA synthesis and transcript accumulation prior to spermiogenesis. Together, our results provide a comprehensive profile of the gene expression program enacted during sperm development, which enables the repression of most housekeeping genes, while driving the specific activation of thousands of spermatogenesis-related coding and non-coding RNAs.

### Gene expression changes are largely transcriptional

To determine to what extent the observed differences in steady-state RNA abundance across spermatogenesis were due to transcription as compared to RNA processing or RNA stability, we directly measured levels of active transcription in purified SG, SC and RS using Precision Run-On sequencing (PRO-seq)[25,26]. This approach allows for high

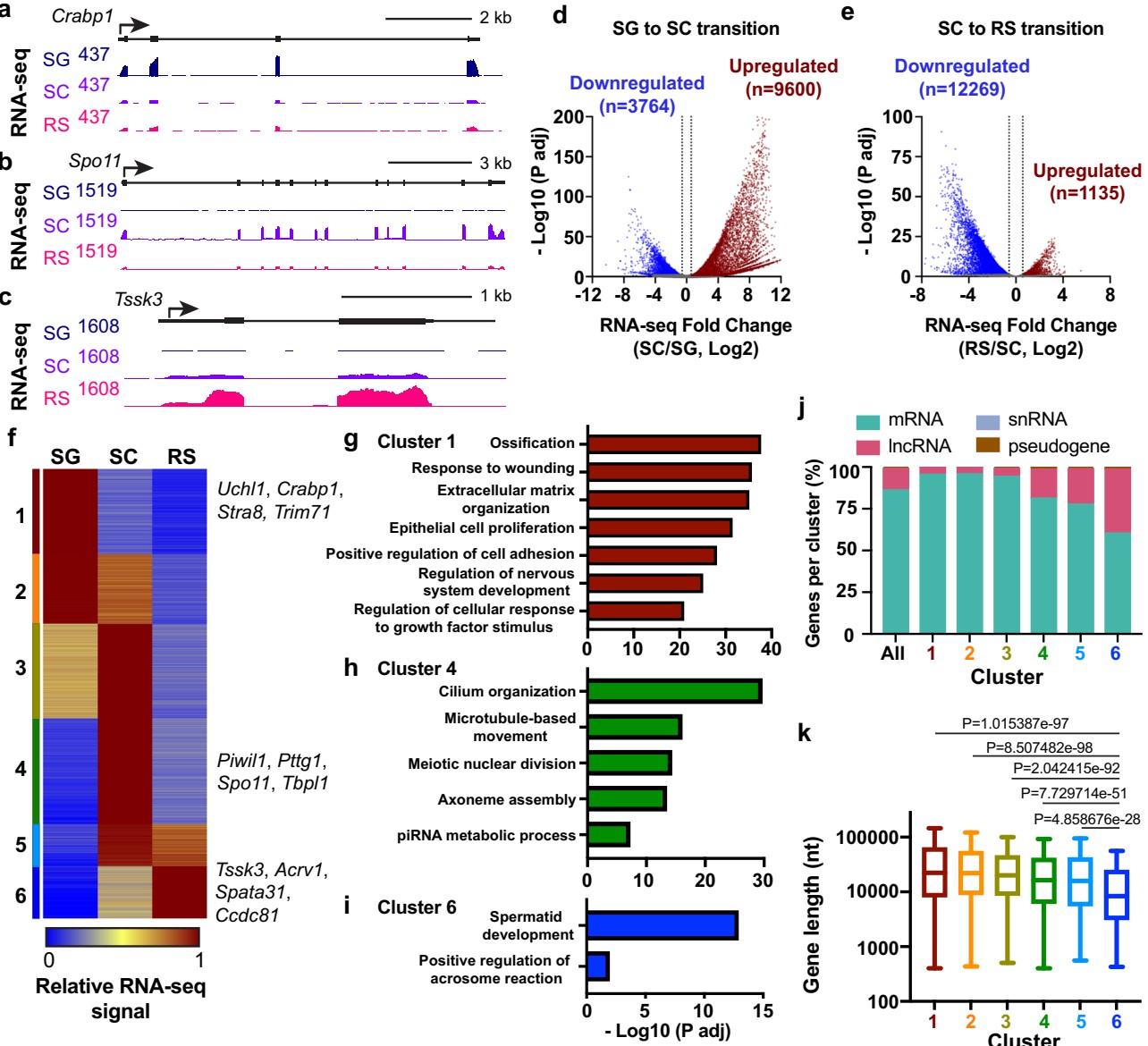

**Fig. 1 | Most active genes are differentially expressed during spermatogenesis.**
**a**–**c** Genome browser views of RNA-seq at *Crabp1*, a SG-specific gene (**a**), *Spo11*, a SC-specific gene (**b**), and *Tssk3*, a RS-specific gene (**c**). **d**, **e** Volcano plots showing fold changes in RNA-seq expression versus adjusted p-values, for active genes ($n = 19,823$). SG compared to SC (**d**), and SC to RS (**e**). Dotted lines indicate 1.5-fold cut-off for significantly increased (red) or decreased (blue) genes, with numbers of genes passing these thresholds and *p*-values < 0.05 indicated on each plot. *P*-values from DESeq2, (Wald test, adjusted p-value calculated using Benjamin-Hochberg correction). **f** Heatmap showing relative RNA-seq expression in each cell type using k-means clustering and standard Euclidean distances. Shown are 17,078 genes differentially expressed in (**d**, **e**). Cluster 1 $n = 3226$, Cluster 2 $n = 2645$, Cluster 3

$n = 3601$, Cluster 4 $n = 4024$, Cluster 5 $n = 1596$, and Cluster 6 $n = 1986$. **g**–**i** Top Gene Ontology terms and significance values for clusters 1 (**g**), 4 (**h**) and 6 (**i**), generated with ClusterProfiler and consolidated using Revigo. Enrichment p-values were calculated by a hypergeometric test, and the p-values were adjusted using a Benjamin-Hochberg correction. **j** Percent of genes from each cluster that correspond to the indicated RNA biotypes, as compared to all genes. **k** Distribution of gene length (TSS to TES) for genes in each cluster defined in (**f**). Line represents median, box represents 25–75th percentile, whiskers represent 1.5X interquartile range. *P*-values from unpaired, two-sided Mann-Whitney U test are shown for cluster 6 vs. all other clusters. Source data are provided as Source Data Fig. 1.

resolution mapping of actively engaged Pol II, with levels of PRO-seq signal within gene bodies (e.g., from 250 nt downstream of the TSS to the TES) providing a reliable measurement of the amount of productive Pol II elongation occurring within each gene (Fig. 2a). As with the RNA-seq samples, we performed spike normalization to enable absolute quantification of PRO-seq signal in each sample.

We first compared the fold changes in RNA-seq vs. PRO-seq signals for each cell transition, to see how often increased or decreased RNA abundance reflected concomitant changes in levels of elongating Pol II. For both the SG to SC transition (Fig. 2b) and the SC to RS transition (Fig. 2c), the data sets are highly correlated, indicating that

changes in transcription drive many of the changes in gene expression during spermatogenesis. We note that the SC to RS transition has both a narrower range of fold changes as well as an overall downward shift in the RNA-seq levels, reflecting a generally lower abundance of RNA in RS. We reasoned that this reflects the haploid status of RS cells, where one fewer genome copy per cell would require twice the density of elongating Pol II to maintain total RNA levels as compared to diploid SC. Notably, this finding implies that gene upregulation in RS, on a per cell basis, would require a dramatic increase in transcription. Indeed, box plots depicting the RNA-seq and PRO-seq fold changes at genes upregulated during the SC to RS transition (Fig. 2d), shows a striking

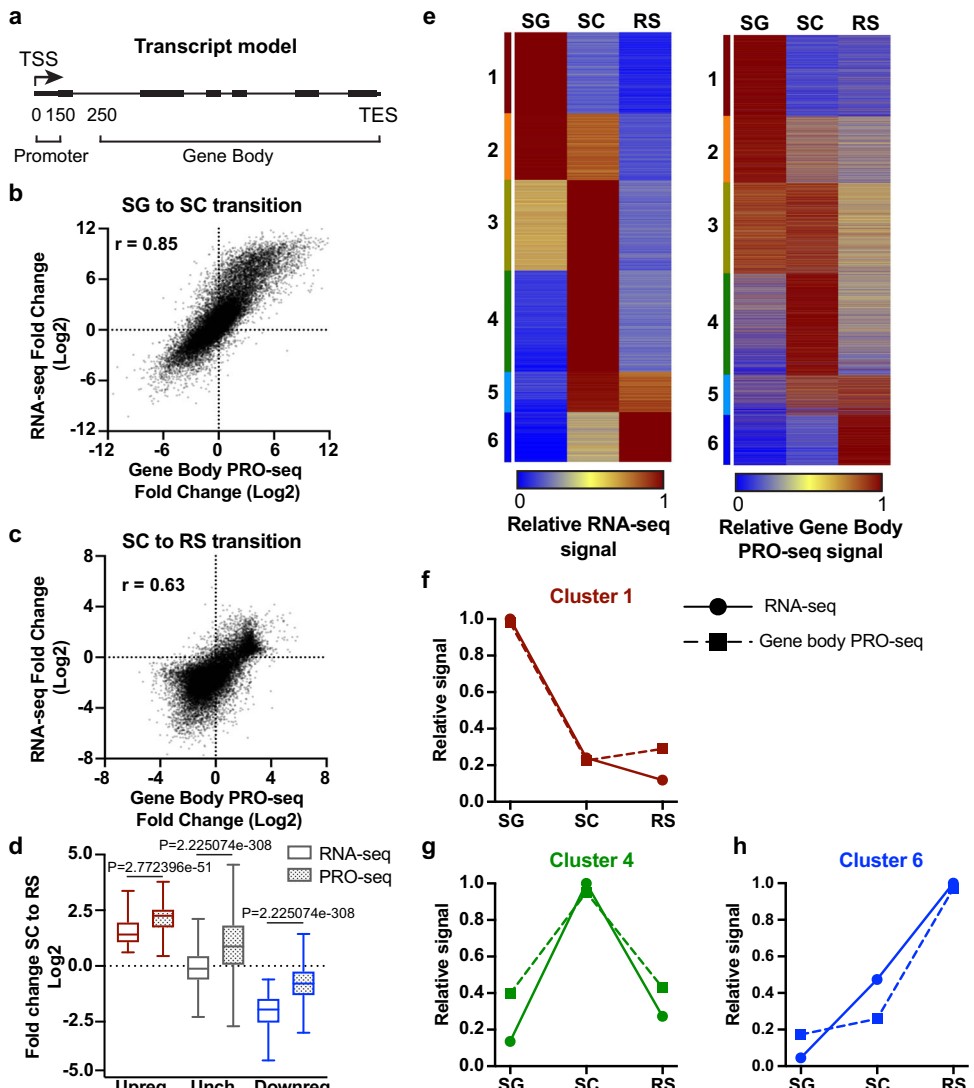

**Fig. 2 | Gene expression changes during spermatogenesis are largely transcriptional. a** Example windows for PRO-seq analysis in promoter (TSS to TSS + 150) or gene body (TSS + 250 to TES). **b** Scatter plot comparing the fold changes between SG and SC in RNA-seq versus PRO-seq gene body signal (TSS + 250 to TES). Spearman's r = 0.85, and $n$ = 19,823. **c** Scatter plot as in **b**, for the fold changes between SC and RS. Spearman's r = 0.63, and $n$ = 19,823. **d** Box plot representation of the distribution of fold changes in RNA-seq (empty box) or PRO-seq gene body (TSS + 250 to TES) (dotted box) for the SC to RS transition. Genes are grouped based on RNA-seq status as upregulated (red, $n$ = 1134), unchanged (gray, $n$ = 5880), or downregulated (blue, $n$ = 12,226) as in Fig. 1e. Line represents median, box represents 25–75th percentile, whiskers represent 1.5X interquartile range. *P*-values from two-sided Wilcoxon signed rank test. **e** Heatmaps show relative RNA-seq (left, as in Fig. 1f) compared to relative PRO-seq gene body (TSS + 250 to TES) signal (right). $n$ = 17,078. **f**–**h** Plots of average relative signal from RNA-seq (solid line and circle) or PRO-seq in gene bodies (dotted line and square) for each cell type, for clusters 1 (**f**), 4 (**g**), and 6 (**h**). Source data are provided as Source Data Fig. 2.

increase in PRO-seq signal at upregulated genes. Importantly, even unchanged genes show evidence of a nearly 2-fold increase in transcription activity in haploid cells as assessed by gene body PRO-seq signal (Fig. 2d, median=0.88 Log2, or a 1.84-fold increase). These findings suggest that, following meiosis, spermatids overcome reduced gene dosage by enacting a form of dosage compensation, which involves ~2-fold activation of genes whose levels should remain consistent between SC and RS.

To further probe the relationship between RNA abundance and transcription levels, we generated a heatmap of relative PRO-seq gene body signal using the same gene list and gene order as shown for the RNA-seq (Fig. 2e: RNA-seq at left, gene body PRO-seq at right). In agreement with transcription being the dominant driver of changes in RNA abundance during spermatogenesis, the RNA-seq and PRO-seq profiles corresponded well (Fig. 2e). Graphing the relative expression levels for RNA-seq and PRO-seq across cell types confirmed the strong

agreement between RNA abundance and active transcription elongation at: Cluster 1 genes that are expressed most highly in SG and repressed in other cell types (Fig. 2f); Cluster 4 genes that are very strongly expressed in SC with lower expression in SG and RS (Fig. 2g); and Cluster 6 genes that are highly active only in the haploid RS stage (Fig. 2h). Agreement between measures of RNA abundance and transcription activity were also observed for genes in clusters 2, 3 and 5 (Supplementary Fig. 2a–c). We conclude that transcription regulation is of central importance in determining RNA abundance in differentiating germ cells.

## SC selective genes accumulate paused Pol II in SG
To dissect the mechanisms of transcription regulation during spermatogenesis, we further analyzed the PRO-seq data. As noted above, after Pol II recruitment to a gene promoter and initiation of RNA synthesis, the polymerase undergoes transient pausing. Paused Pol II

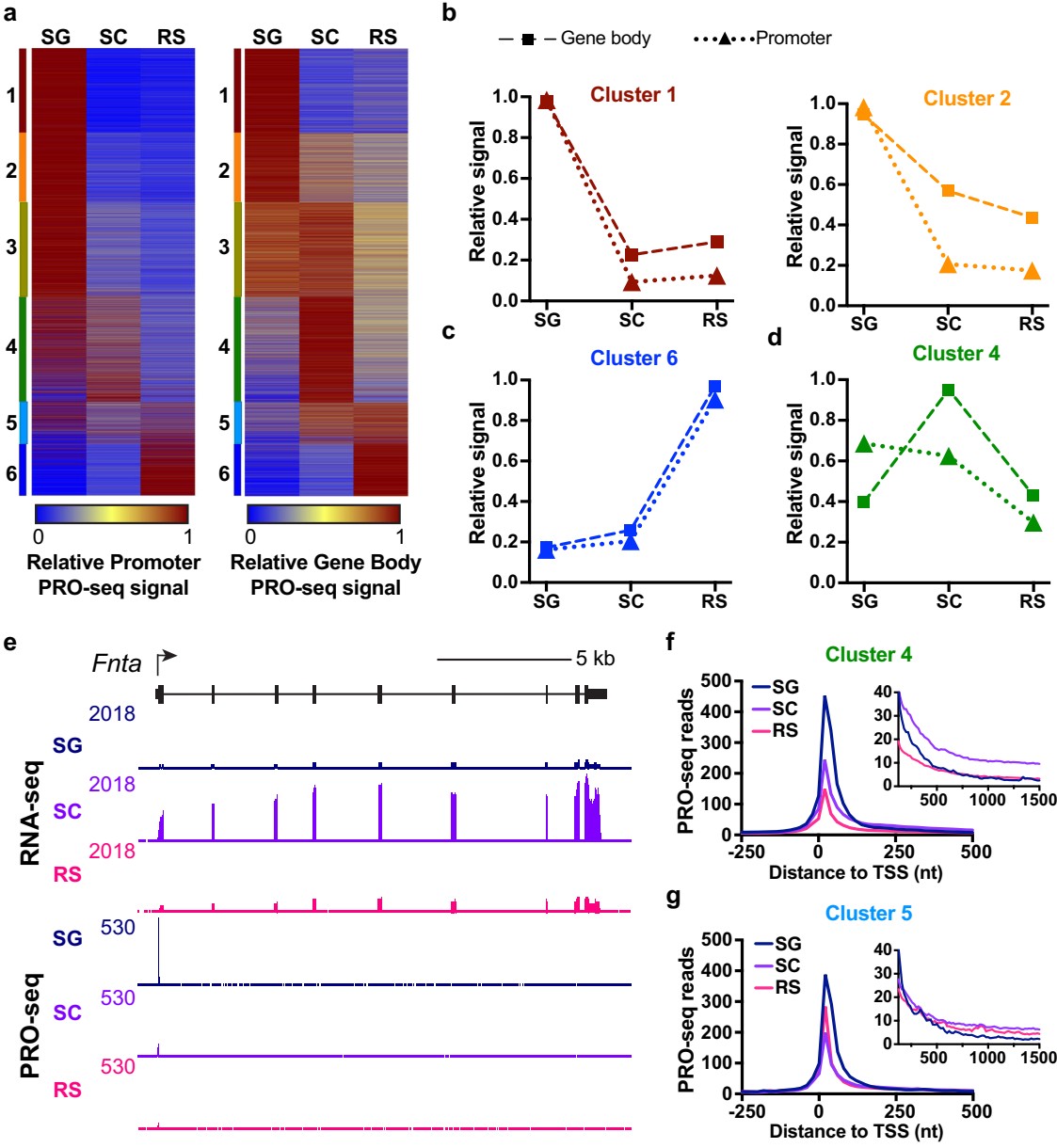

**Fig. 3 | Analysis of promoter PRO-seq signal reveals that SC selective genes accumulate paused Pol II in SG. a** Heatmaps show the relative promoter PRO-seq signal (TSS to +150, left) as compared with the gene body PRO-seq signal (at right, as in Fig. 2e, right). Genes are ranked as in Figs. 1f, 2e. **b–d** Plots of average relative signal from PRO-seq within promoter (dotted line with triangles) or gene body (dashed line with squares) regions for each cell type. Shown are data for genes in clusters 1 and 2 (**b**) cluster 6 (**c**) or cluster 4 (**d**). **e** Genome browser view of RNA-seq and PRO-seq at *Fnta*, a SC-specific cluster 4 gene that has a prominent peak of promoter PRO-seq in SG. **f–g** Metagene plots of average PRO-seq reads in each cell type for all protein-coding genes in cluster 4 (**f**) or cluster 5 (**g**). Inset y-axis is zoomed-in to show gene body signal from +125 to +1500 nt downstream of the TSS. Source data are provided as Source Data Fig. 3.

remains transcriptionally engaged while bound by the DSIF and NELF complexes, which stabilize the paused state and disfavor forward synthesis[27–30]. Pause release is triggered by the recruitment of the kinase P-TEFb, which phosphorylates Pol II and DSIF to dissociate NELF and enable the transition to productive elongation. Importantly, loss of NELF renders the early elongation complex susceptible to premature termination and loss of gene activity[31–33], highlighting the importance of this checkpoint prior to Pol II release into the gene body.

To investigate a role for Pol II pausing at genes regulated during spermatogenesis, we generated a heatmap of relative PRO-seq signal at promoters (Fig. 3a, left), which represents the levels of initiated and promoter-proximally paused Pol II. We then compared this heatmap to that depicting the relative PRO-seq signal within gene bodies (Fig. 3a,

right; as shown in Fig. 2e, right), which represents the level of active RNA synthesis. If, for example, reduced transcription levels in a given cell type result from a decrease in Pol II initiation at the gene promoter, then both the promoter and gene body PRO-seq signals should be lower in that cell type. This scenario is reflected at genes in clusters 1 and 2, wherein the promoter and gene body PRO-seq signals are highest in SG and are sharply reduced in SC and RS (compare Fig. 3a left and right). Cluster 1 genes in particular show remarkably similar profiles of promoter and gene body signal across cell types (Fig. 3b), indicating that the decrease in abundance of cluster 1 transcripts results from suppression of transcription initiation in SC. Similarly, we find that cluster 6 genes are primarily regulated at the level of transcription initiation (Fig. 3c), where increased expression in RS reflects higher average levels of PRO-seq at both the promoters and within

gene bodies. In agreement with these observations, we find that the core promoter motif for the TATA-binding protein (TBP) is enriched at cluster 1 and cluster 6 genes (Supplementary Fig. 3a). TBP is a central factor in transcription initiation, and high levels of TBP and TBP-like factor (TLF) have been observed during spermatogenesis, with TLF particularly enriched and required for RS function[34,35].

By contrast, genes in clusters 3–5 show highest promoter PRO-seq signal in SG, despite reaching maximal expression at the SC or RS stage (Fig. 3a, d, Supplementary Fig. 3b, c). This profile is consistent with promoter-proximal pausing at these genes in SG, potentially poising them for expression later in development. Indeed, the sharp increase in gene body polymerase signal observed at cluster 4 genes in SC is accompanied by a drop in promoter Pol II occupancy in this cell type (Fig. 3d and Supplementary Fig. 3d, e), implying that higher level expression results from faster pause release (Fig. 3d). Inspection of example cluster 4 genes in the genome browser supports this conclusion (Fig. 3e, e.g., *Fnta*): these genes exhibit highest RNA-seq signal in SC, but promoter PRO-seq levels are highest in SG, prior to the onset of gene activity. A metagene profile of PRO-seq reads at genes in cluster 4 further emphasizes this point: PRO-seq signal is highest at the TSS in SG, but significant PRO-seq signal is only observed within the gene body in SC (Fig. 3f, see inset). Similarly, metagene plots of genes in cluster 5, which are activated in SC and remain modestly expressed in RS, display highest promoter PRO-seq signal in SG (Fig. 3g). Cluster 3 genes (Supplementary Fig. 3f) are likewise activated in SC by markedly increased pause release. We conclude that genes activated during the transition from SG to SC are poised for activation in the earlier SG stage, and that gene induction involves stimulation of pause release.

## NELF-B is required for the completion of spermatogenesis

To directly evaluate a role for Pol II pausing in spermatogenesis, we generated conditional knockout (cKO) mice that lacked the pause-inducing factor NELF-B in male germ cells by crossing the *Stra8-iCre* mouse strain with a NELF-B floxed mouse strain reported previously[6,36]. Notably, the NELF subunits are interdependent for stability, such that depletion of NELF-B disrupts NELF complex formation and prevents activity. *Stra8-iCre* mediated excision of floxed genes takes place in SG of the testis starting from postnatal day 4 (PND4), so that all subsequent male germ cell types including SC, RS, and elongated spermatozoa would be null for NELF-B. However, we note that *Stra8-iCre* mediated excision reaches full penetrance between PND21-40[37].

First, to assess the levels of NELF-B in control versus NELF-B cKO mice, immunohistochemistry was performed in testis cross sections. This confirmed depletion of NELF-B signal in male germ cells at both PND15 and 24 (Fig. 4a–d, example germ cells denoted with G). As anticipated, the somatic Sertoli cells of the testis continued to express NELF-B (Fig. 4a–d, Sertoli cells denoted with S). To determine whether loss of NELF-B impacts spermatogenesis, we examined testis histology from PND15, PND24, PND35, and adult mice (Fig. 4e–h and Supplementary Fig. 4a–d). Vacuole formation and disorganization of meiotic cells were evident in NELF-B cKO testis by PND15 (Fig. 4e, f). At PND24, the control mice showed the presence of RS in a portion of the tubules, as expected (Fig. 4g). In contrast, the cKO testis showed further loss of germ cells and increased appearance of vacuoles (Fig. 4h). The severity of the phenotype worsened with age. By PND35 the number of tubules with absent or disorganized germ cells increased (Supplementary Fig. 4a, b). *Stra8-iCre* mediated excision starts in the undifferentiated SG on PND4 but reaches full penetrance by PND40[37]. Thus, complete germ cell depletion in the testes of PND35 NELF-B cKO mice likely reflects the requirement of NELF-B for spermatogonial stem cell survival/differentiation. The 9-month-old testes from NELF-B cKO mice showed tubules devoid of any germ cells, indicating that there was no recovery of spermatogenesis (Supplementary Fig. 4c, d). Together, these results indicate that loss of NELF-B disrupts the normal progression through spermatogenesis. Overall, these defects were

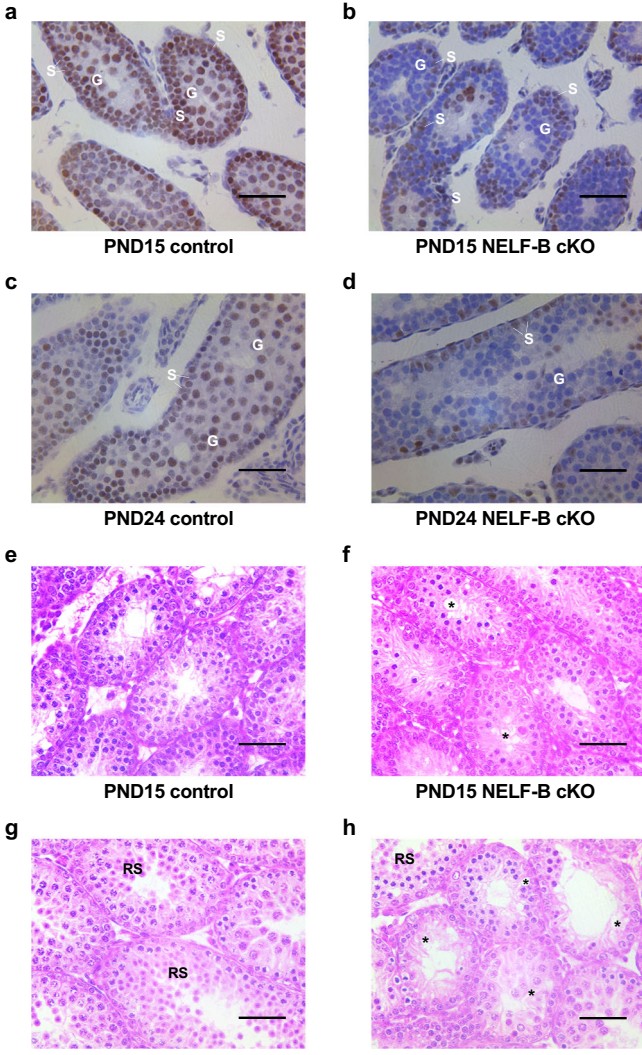

**Fig. 4 | NELF-B is required for the successful completion of spermatogenesis.** **a–d** Immunohistochemistry using NELF-B antibody stains NELF-B-positive Sertoli (S) and germ cells (G) in testis from control PND15 (**a**) and PND24 (**c**) mice. Quantification confirms 100% positive staining in $n = 50$ tubules from mice at PND15 and PND24. In tubules from NELF-B cKO mice at PND15 (**b**) and PND24 (**d**), the germ cells (G) are devoid of NELF-B protein while the Sertoli (S) cells remain NELF-B positive. Quantification identified $n = 10$ out of $n = 72$ tubules (13.89% positive) at PND15, and $n = 8$ out of $n = 50$ tubules (16% positive) at PND24. Scale bar, 50 μm. We note that some NELF-B positive germ cells are detected in the NELF-B cKO, which represent cells that escaped Cre-mediated excision of NELF-B. **e–h** Testis cross sections of PND15 and PND24 control ($n = 3$, **e**, **g**) or NELF-B cKO ($n = 3$, **f**, **h**) mice. PND15 NELF-B cKO mice showed degeneration of the epithelium (asterisk) involving meiotic cells in some areas (**f**). At PND24, vacuoles are also formed, and the epithelium is disorganized with detached and fewer RS (**h**). Scale bar, 50 μm.

reminiscent of our recent work using TDP-43 germ cell cKO[6], suggesting that there could be similarities in the defects caused by these two regulators of early transcription elongation and gene expression.

## NELF-B and TDP-43 are required for germ cell maturation

We next sought to investigate transcriptional changes in germ cells lacking NELF-B or TDP-43. Our prior studies of a germ cell specific TDP-43 cKO, and our current data on the NELF-B cKO (Fig. 4) demonstrate that loss of either of these factors disrupts germ cell development, making it difficult to obtain sufficient SC or RS cells for bulk RNA-seq or PRO-seq experiments. Thus, we performed 10X single cell RNA-seq (scRNA-seq) on whole testis samples from PND24 mice. At this

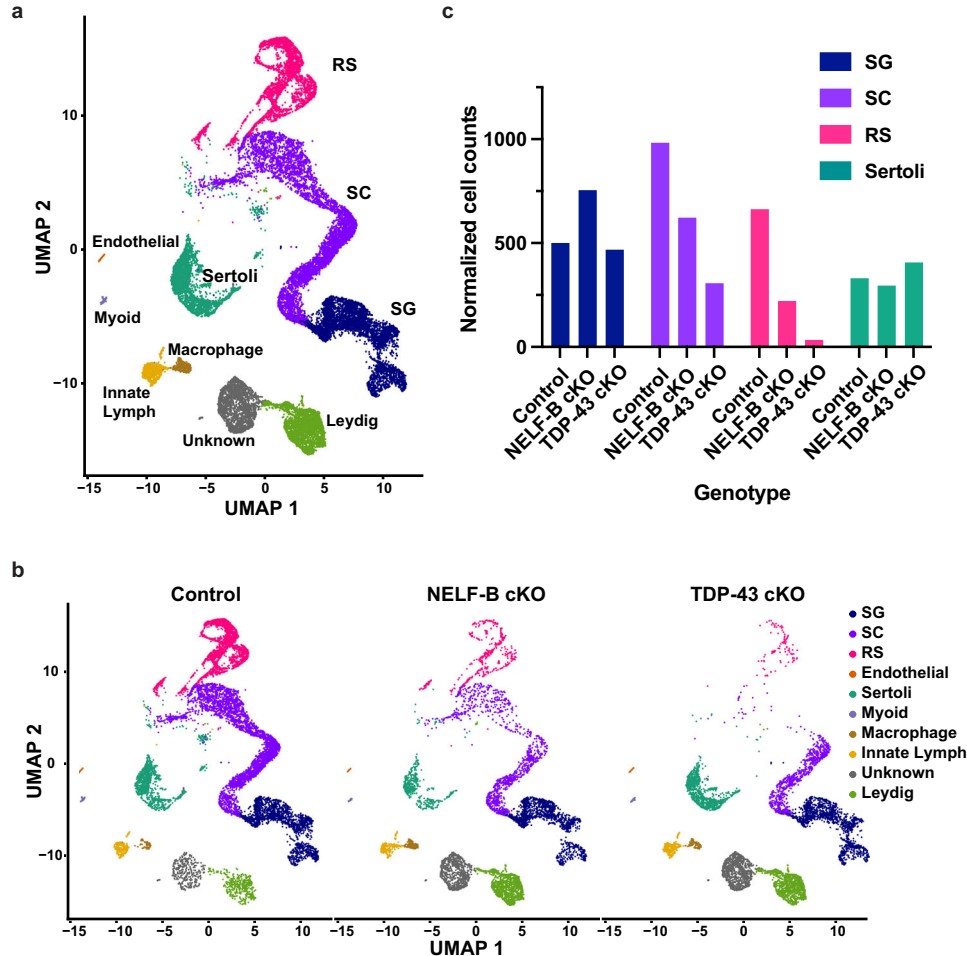

**Fig. 5 | NELF-B and TDP-43 are required for the proper maturation of germ cells.** **a** UMAP representation of all 10 somatic and germ cell type clusters, integrated across the 3 genotypes ($n = 24,269$ cells). **b** Cell clusters from (**a**), separated by genotype. (Control: $n = 11,170$, NELF-B cKO: $n = 6404$, TDP-43 cKO: $n = 6695$ cells). **c** Numbers of each germ cell type and somatic Sertoli cells per genotype. Cell counts were normalized by the number of mice per genotype (Control: $n = 4$, NELF-B cKO: $n = 2$, TDP-43 cKO: $n = 3$). Source data are provided as Source Data Fig. 5.

developmental stage, we observe defects histologically in cKO animals (Fig. 4), but all three germ cell types (SG, SC, RS) are present.

We detected 24,269 total cells across the three genotypes. These cells were distributed across both germ and somatic cell types, with each cell having a median of 12,130 unique molecular identifiers (UMIs) and 4285 genes identified. Clustering of the cells using previously defined marker genes led to the distinction of the 10 cell types (Fig. 5a, Supplementary Fig. 5a) observed in an earlier study of mouse spermatogenesis[11]. The only cell type found previously that we did not also computationally distinguish was elongating spermatids, because PND24 is too early for the first wave of spermatogenesis to yield elongated spermatozoa.

To define the effects of NELF-B or TDP-43 loss on germ cell development, the data were separated by genotype. In control mice, expression of both NELF-B and TDP-43 was high in SG and SC as compared to other cell types (Supplementary Fig. 5b), consistent with roles early in spermatogenesis. Analysis of cell type distribution across genotypes revealed substantial losses in representation of SC and RS among cells from the NELF-B ($n = 2$) and TDP-43 ($n = 3$) cKO lines compared to control mice ($n = 4$) (Fig. 5b). In contrast, the cluster representing SG and most of the somatic cell types were not overtly affected by loss of either NELF-B or TDP-43. These observations were quantified by comparing cell counts, normalized by the number of mice per genotype, across the germ cell types and the somatic Sertoli cells (Fig. 5c).

We next examined which stage of prophase I was susceptible to loss of NELF-B and TDP-43. Using previously defined marker genes, we separated the SC cluster into preleptotene, leptotene, zygotene, pachytene, and diplotene sub-clusters (Supplementary Fig. 5c)[38,39]. Although the impact of NELF-B or TDP-43 loss was apparent beginning from the zygotene stage, loss of NELF-B affected prophase I gradually as noted by the persistence of some pachytene and diplotene SC (Supplementary Fig. 5d). In contrast, loss of TDP-43 severely affected the pachytene stage, consistent with the high level of expression of TDP-43 in pachytene SC and its requirement for proper synapsis and homologous recombination[6]. We conclude that loss of NELF-B causes significant defects in male germ cell maturation, and that developmental defects are further exacerbated in TDP-43 cKO animals.

## NELF-B and TDP-43 regulate genes critical for spermatogenesis

In light of the substantial losses of differentiated pachytene, diplotene SC and RS cells in the NELF-B and TDP-43 cKO mice (Fig. 5b, Supplementary Fig. 5d), we wanted to understand which genes were misregulated in SG and early meiotic SC cells (preleptotene, leptotene, and zygotene) to prevent proper germ cell maturation. We thus ran a pseudobulk differential expression analysis on cells within the SG and early meiotic clusters, comparing both cKO genotypes to WT control. This identified 207 differentially expressed genes in NELF-B cKO, which were evenly distributed between up- and downregulation (Fig. 6a). A similar analysis found 842 differentially expressed genes for TDP-43

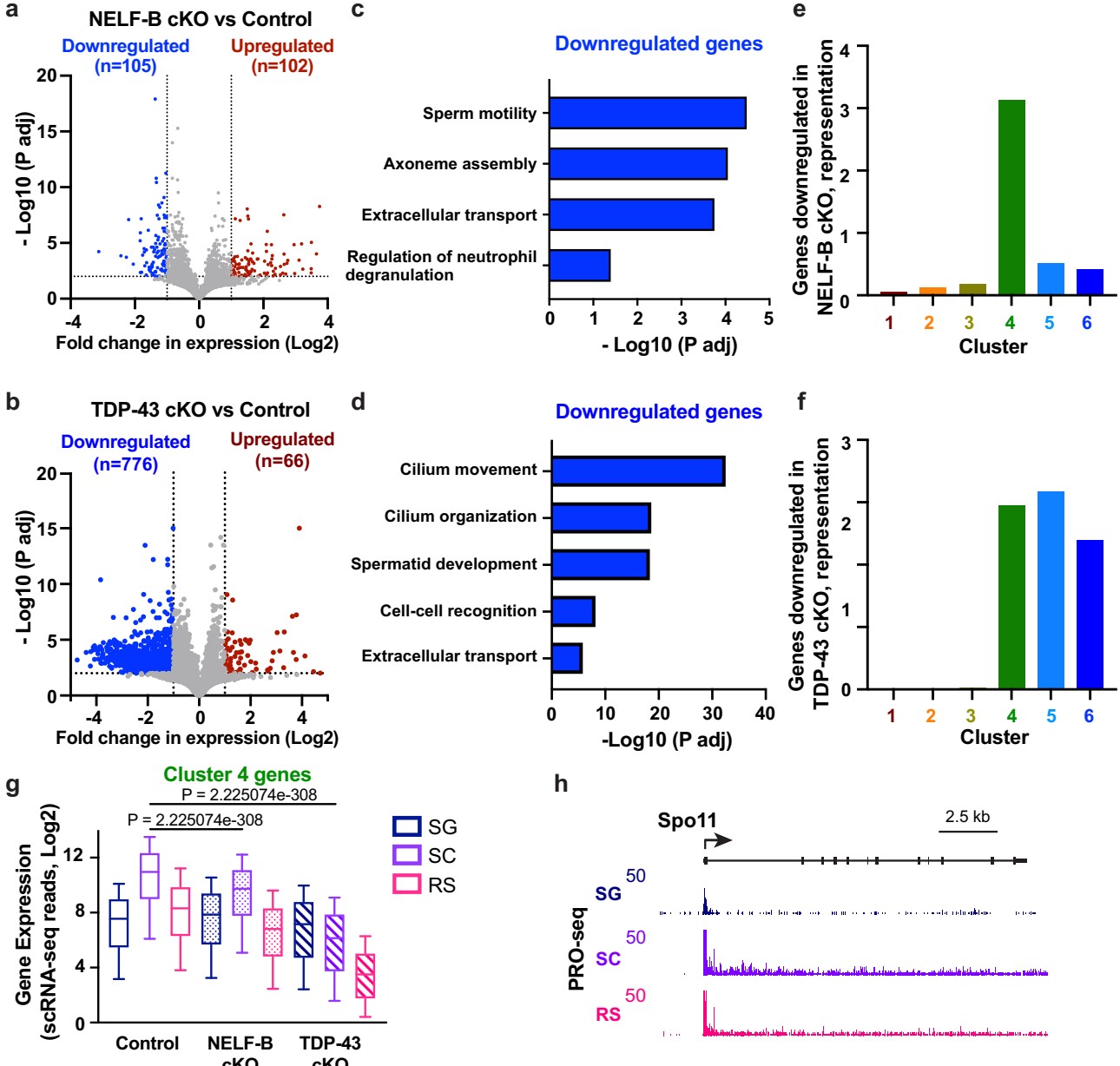

**Fig. 6 | Genes affected in the absence of NELF-B or TDP-43 are critical for SC and RS development. a, b** Volcano plots of fold changes in gene expression in SG and early meiotic cells (pL, L &Z) versus adjusted p-values, for control mice compared to NELF-B cKO (**a**) or TDP-43 cKO (**e**) mice. Dotted lines indicate 2-fold cut-off for significantly increased (red) or decreased (blue) genes, with numbers of genes passing these thresholds and *p*-values < 0.01 indicated on each plot. *P*-values from DESeq2, (Wald test adjusted p-value calculated using Benjamin-Hochberg correction). **c, d** Enriched functional categories among downregulated genes in NELF-B (**c**) or TDP-43 (**d**) mice, and their significance determined by ClusterProfiler. Enrichment *p*-values were calculated by a hypergeometric test, and the p-values were adjusted using a Benjamin-Hochberg correction. **e, f** Representation of downregulated genes from NELF-B cKO (**e**) or TDP-43 cKO (**f**) mice, within each cluster (from Fig. 1f). A value of 1 indicates representation expected by chance. **g** Expression of Cluster 4 genes (*n* = 3916) in scRNA-seq from control, NELF-B cKO, or TDP-43 cKO mice, in indicated cell types. Line represents median, box represents 25–75th percentile, whiskers represent 10–90th percentile. *P*-values from two-sided Wilcoxon signed rank test. **h** Genome browser view of wild type PRO-seq signal in each cell type at *Spo11*. Note that the y axis is truncated at 50 to highlight PRO-seq gene body signal. Source data are provided as Source Data Fig. 6.

cKO, which were strongly skewed towards downregulation (Fig. 6b). These data are consistent with current models for TDP-43 function in increasing gene activity, at the level of transcription or RNA stability.

To help interpret these gene sets, we investigated their functional enrichment. We observed no significantly enriched gene ontology categories or pathways for the upregulated genes from either genotype. We thus focused on the downregulated genes. For both genotypes, the most enriched functional categories were related to spermatid development and motility (Fig. 6c, d). This implies that the

absence of NELF-B or TDP-43 leads to the reduced expression of genes critical for normal spermatogenesis. Given the role of the NELF complex in stabilizing promoter proximal pausing, we wondered if the genes affected by NELF-B cKO might be those normally occupied by paused Pol II in these cell types. In particular, we hypothesized that pausing mediated by NELF in SG might be important to promote open promoter chromatin and poise genes for further activation in SC (e.g., cluster 4 genes, Fig. 3d–f). To test this idea, we investigated the downregulated genes from the single cell analyses with respect to the

gene clusters identified in the bulk analyses (Fig. 1f). Indeed, genes downregulated in NELF-B cKO were significantly overrepresented by cluster 4 genes (Fig. 6e), with other clusters under-represented. We conclude that the genes most affected by NELF-B cKO are those that are highly paused in SG, and suggest that pausing at these genes in SG primes them for robust activation as germ cells differentiate towards SC. To directly test whether Cluster 4 genes are broadly affected by loss of NELF-B, we evaluated expression levels of all Cluster 4 genes in scRNA-seq data from each genotype (Fig. 6g). These results confirm that the activation of Cluster 4 genes that normally occurs in SC is dramatically muted in NELF-B cKO animals. Notably, expression of Cluster 4 genes is also repressed in TDP-43 cKO SC.

Analysis of downregulated genes in TDP-43 cKO SG and early meiotic cells revealed an enrichment of genes expressed in both SC and RS (Fig. 6f, enrichment within clusters 4–6). These data suggest that TDP-43 supports the low-level expression of genes in SG, perhaps to enable their activation in either SC or RS (see Fig. 6g for the effect of TDP-43 cKO on cluster 4 genes). By contrast, genes upregulated in NELF-B or TDP-43 cKO cells are over-represented in cluster 1 genes (Supplementary Fig. 6a, b), suggesting that germ cells lacking NELF-B or TDP-43 become fixed into an earlier stage of development and fail to progress through spermatogenesis. Given the overlapping gene ontology terms observed in NELF-B and TDP-43 cKO samples, and similar defects in spermatogenesis, we directly compared genes downregulated in each genotype. Of the 105 genes downregulated upon loss of NELF-B, 53 are also downregulated upon loss of TDP-43 (Supplementary Fig. 6c), representing a highly significant overlap. A similar comparison of upregulated genes also supported overlapping functions of NELF-B and TDP-43 (Supplementary Fig. 6d).

One of the shared downregulated genes was *Spo11*, an SC marker gene that is critical for meiosis (Fig. 1b). Notably, while *Spo11* is a cluster 4 gene that is maximally induced in SC, we observe paused Pol II at the *Spo11* promoter in SG from control animals and observe a detectable level of 'priming' transcription in SG cells (Fig. 6h). Given the enrichment of genes involved in meiosis in Cluster 4, we then specifically looked at the expression of Cluster 4 genes associated with meiotic nuclear division (Supplementary Fig. 6e, *n* = 91 genes). Indeed, NELF-B cKO animals showed significantly less activation of these meiotic genes than control animals. Further, this meiotic gene set was downregulated in SC of animals with TDP-43 cKO, consistent with our previous study showing that cKO of TDP-43 in SG lead to meiotic arrest at mid-pachytene stage with synapsis defects[6]. Taken together, these data demonstrate that NELF-B and TDP-43 are important for appropriate gene expression in spermatogenesis, with particular defects observed at genes with critical roles in the onset and progression of meiosis, such as *Spata22, Spo11, Meiob,* and *Rad51c* (Supplementary Table 1).

## Promoter activity in SG correlates with sites of DSBs in SC

Given the dependence of germ cell maturation on NELF-B as well as TDP-43, and of *Spo11* as a shared target of these factors, we asked whether NELF-mediated pausing might affect meiosis. During prophase I of meiosis, double-strand breaks (DSBs) are formed by the topoisomerase SPO11 in leptotene SC[40,41]. The repair of these breaks by homologous recombination and synapsis begins in zygotene SC and is completed by the pachytene stage. These events play important roles in the segregation of homologous chromosomes. It is believed that approximately 300 DSBs are formed per SC, but what determines the choice of DSB sites remains unclear. Recent sequencing of SPO11 associated DNA oligos has elucidated the sites of DSBs in mouse SC, revealing that DSBs are enriched in the vicinity of nucleosomes modified on histone H3 by Lysine 4 trimethylation (H3K4me3)[40,41]. Although the targeting of SPO11 is not thought to be dependent on H3K4me3, DSB formation appears to require H3K4me3 deposited by the methyltransferase PRDM9[42]. Given the strong connection between levels of H3K4me3-modified nucleosomes and paused Pol II at gene

promoters[43], we wished to investigate whether DSB formation might occur proximal to paused polymerases, and whether the disruption of pausing might perturb meiosis.

Using SPO11-oligo data in SC, generated previously to map DSBs in mice of similar genotypes to our C57BL/6 J model[40], we began by examining SPO11-oligo reads near TSSs that are active and differentially expressed in our RNA-seq analyses (*n* = 17,078, as in Fig. 1f). Composite metagene plots of SPO11-oligo reads revealed a peak in signal centered just downstream of the TSS (Fig. 7a), in a location coincident with that of paused Pol II and the H3K4me3-modified +1 nucleosome. To further connect the DSB sites to transcriptional activity we asked how levels of DSBs, as assessed by SPO11-oligo reads, correlated with read counts from H3K4me3 ChIP-seq[44], or PRO-seq (Supplementary Fig. 7a), considering separately PRO-seq reads at gene promoters (representing paused Pol II) and those within gene bodies (representing productive transcription elongation). Interestingly, the strongest relationship with SPO11-oligo signal was observed with promoter-proximal PRO-seq reads in SG (Supplementary Fig. 7a, Spearman's rho = 0.44). Indeed, if we rank genes by promoter PRO-seq signal in SG and generate heatmaps of SG PRO-seq and SPO11-oligo data, there is a clear agreement between these data sets (Fig. 7b). Genes in the top quartile of promoter PRO-seq in SG showed significantly more SPO11-oligo signal at or around the TSS (Fig. 7c) than do genes in the bottom quartile of promoter PRO-seq read counts (Fig. 7d). This trend was consistent across all quartiles ranked by promoter PRO-seq signal, with decreasing PRO-seq reads corresponding to reduced levels of DSBs at these promoters (Fig. 7e). These results suggest a relationship between paused Pol II in SG and DSB formation in the subsequent SC stage. Notably, DSB formation near promoters in SC would be concomitant with Pol II release from promoter regions and gene activation.

Given the strong connection between NELF-mediated pausing in SG and the location of DSBs, we investigated progression through meiosis in our NELF-B cKO mutant mice. As noted above, scRNA-seq indicated a significant drop in SC number in NELF-B cKO mice compared to WT control (Fig. 5c). H&E-stained testis cross sections of PND24 mice were evaluated for SC at different stages of meiosis using morphological criteria. In control mice we observe leptotene and zygotene, as well as pachytene SC indicating progression of meiosis (Fig. 7f). In contrast, NELF-B cKO mice produce significantly less pachytene stage SC, indicating failure to complete prophase I of meiosis (Fig. 7f, quantified in Fig. 7g). Further, visualization of an acrosomal marker showed loss of RS in NELF-B cKO mice, further substantiating meiotic failure (Supplementary Fig. 7b, c). This agrees with the scRNA-seq data showing a reduction in pachytene SC and RS in NELF-B cKO animals (Supplementary Fig. 5d).

To further characterize the meiotic defect, we hypothesized that the diminished transcription of genes including *Spo11, Meiob, and Rad51c* in NELF-B cKO cells would lead to a disruption in DSB formation and/or repair. Immunostaining of meiotic spreads with a DSB marker, γ-H2AX (Fig. 8a) showed signal restricted to the sex/XY-body of pachytene SC in the wild-type, as expected. In contrast, in NELF-B cKO pachytene-like SC the γ-H2AX signal persisted along the autosomes (Fig. 8a, lower panel) suggesting a delay in DSB formation or disruption in its repair process. Quantification showed that γ-H2AX signal occupied a much broader nuclear area in NELF-B cKO compared to the wild-type control (*p* < 0.0001) (Fig. 8a, right) indicating impaired DSB repair. This finding is consistent with the defective activation of numerous DSB-repair and meiosis-related genes in NELF-B cKO (Supplementary Fig. 6e, Supplemental Table 1). Furthermore, we observed that replication protein A (RPA) foci persisted in pachytene-like SC of NELF-B cKO mice (Fig. 8b), indicating that DSB sites remain unrepaired. Quantification revealed a significant increase in RPA foci in pachytene-like SC of NELF-B cKO (*n* = 46) compared to the control (*n* = 51) (*p* < 0.0001) (Fig. 8b, right). We suggest that the dysregulation

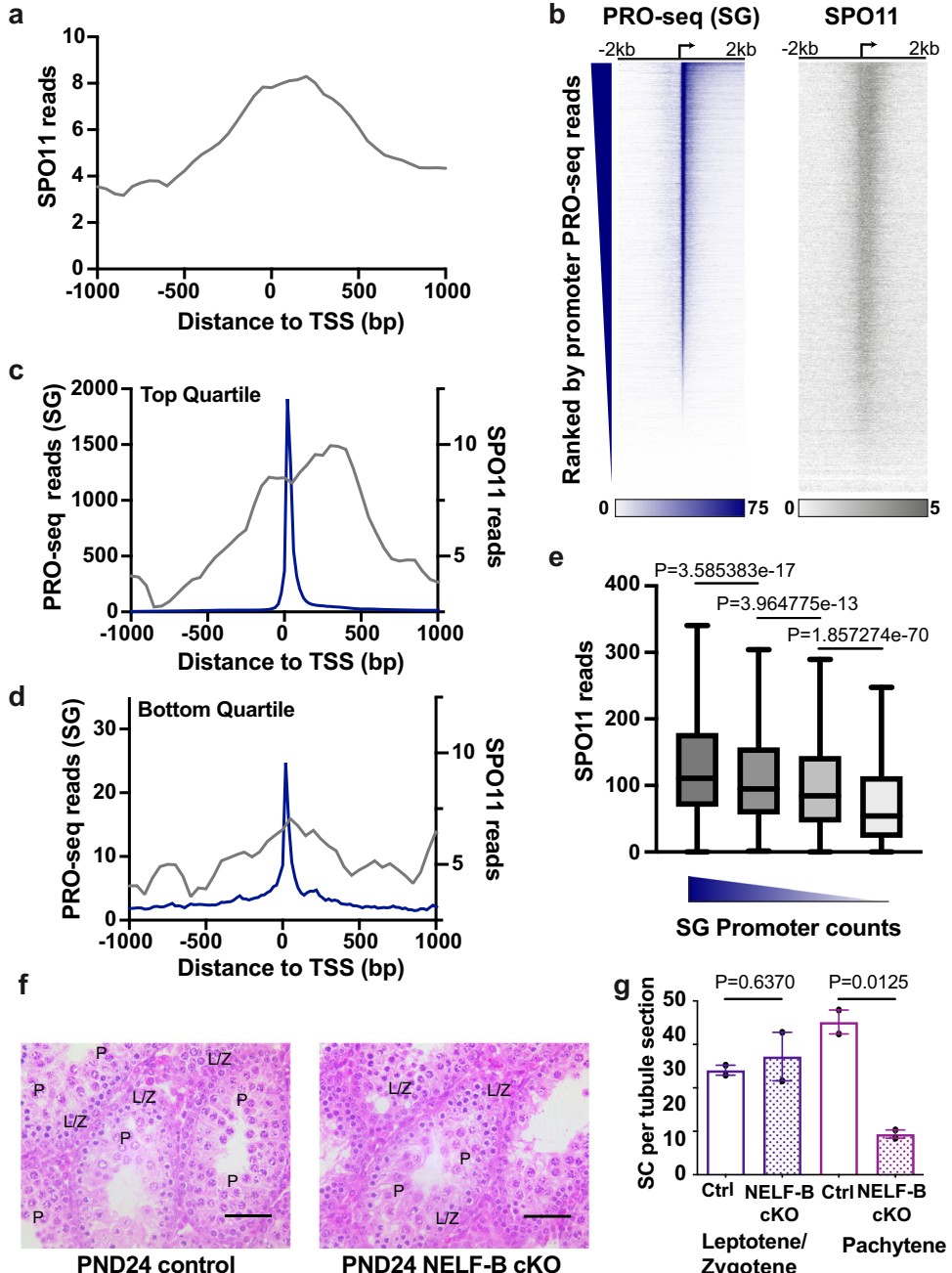

**Fig. 7 | Promoter activity in SG correlates with sites of double-strand breaks in SC. a** Metagene plots of SPO11-oligo reads around the TSS for all DE genes (*n* = 17,078). **b** Heatmaps of SG PRO-seq (left, dark blue) or SPO11 oligo reads (right, gray), aligned at TSSs (indicated by arrow) with genes ranked by SG promoter PRO-seq signal (TSS to TSS + 150, *n* = 17,078). **c, d** Metagene plots around the TSSs for the top (**c**) or bottom (**d**) quartile based on SG PRO-seq reads in promoters with at least 5 reads (*n* = 14,804, with *n* = 3701 genes per quartile). PRO-seq signal for SG (blue) is plotted on the left *y*-axis, SPO11 oligo reads (gray) is plotted with the right *y*-axis. **e** Box plots of the distribution of SPO11 reads at each quartile (*n* = 3701) of SG PRO-seq reads at promoters with at least 5 reads (*n* = 14,804). Line represents the median, whiskers represent 1.5X interquartile range. *P*-values from two-sided Wilcoxon signed rank test. **f** Testis cross sections of PND24 control (*n* = 2, left) or NELF-B cKO (*n* = 2, right) mice. L/Z, indicates SC in Leptotene/Zygotene stages, and P, indicates pachytene SCs. Scale bar, 50 µm. **g** Quantification of SC at Leptotene/Zygotene versus Pachytene stages per tubule. Shown are the mean and range of *n* = 2 biologically independent animals. *P*-value is from Unpaired two-sided t-test. Source data are provided as Source Data Fig. 7.

of pausing in NELF-B cKO SG perturbs activation of genes involved in DSB formation and repair (e.g., *Spo11, Rad51c, Meiob*) thereby impairing DSB formation/repair and proper progression of meiosis.

## Discussion

This work establishes a central role for Pol II pausing in spermatogenesis, demonstrating that male germ-cell specific deletion of the pause-inducing factor NELF-B prevents the progression of SC through

differentiation and meiosis. Specifically, we find that pausing is required in SG to poise genes for activation in the SC state. These findings expand upon a recent study focused on mouse SC which reported pause release in the pachynema stage of SC, largely through recruitment of BRDT by the A-MYB protein[45]. By investigating gene expression across spermatogenesis, we show that paused Pol II is established at thousands of cluster 4 and 5 genes in SG, which undergo coordinated pause release as cells progress through the stages of SC,

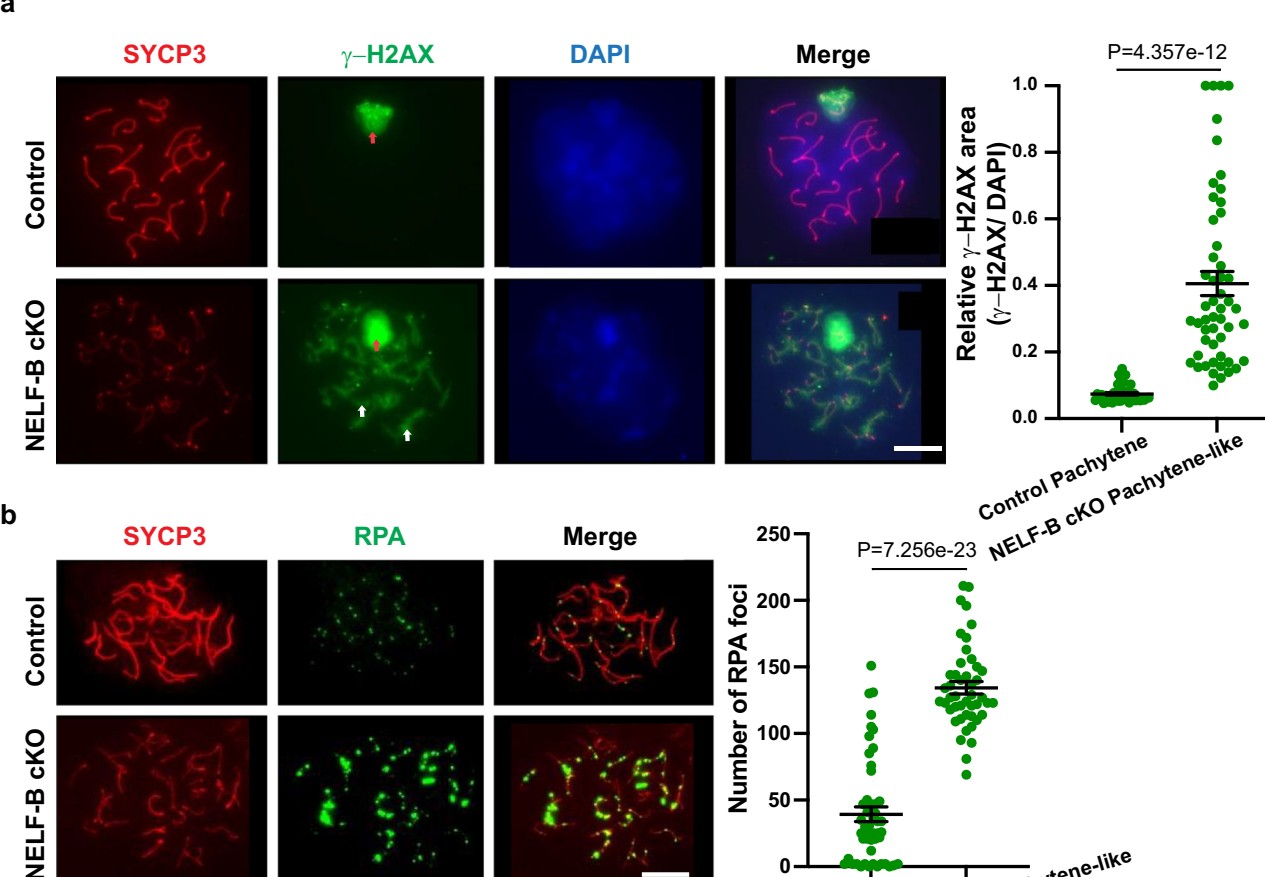

**Fig. 8 | NELF-B cKO mice have impaired DSB repair. a** SC showing SYCP3 (red), γ-H2AX (green) and DAPI (blue) in control (top row) and NELF-B cKO (bottom row). γ-H2AX signal is restricted to the sex body (red arrow) in control whereas it persists on autosomes (white arrows) in NELF-B cKO indicating impairment of DSB repair. Scale bars = 20 µm. At right: Quantification of relative γ-H2AX area (γ-H2AX/DAPI) between the control ($n = 40$) and NELF-B cKO ($n = 52$) from three biological replicates. Mean indicated with horizontal line. Error bars represent standard error of the mean. *P*-value from two-sided *t*-test. **b** SC showing SYCP3 (green) and RPA (red) in control (top row) and NELF-B cKO (bottom row); the higher number of RPA foci in NELF-B cKO implies impaired DSB repair. Scale bars = 20 µm. At right: Quantification of the RPA foci numbers at pachytene from control ($n = 51$) and NELF-B cKO ($n = 46$) from three biological replicates. Mean indicated with horizontal line. Error bars represent standard error of the mean. P-value from two-sided t-test. Source data are provided as Source Data Fig. 8.

and RS, respectively. We therefore suggest the involvement of several pause release factors that function at distinct steps in sperm development, to enable the timely expression of genes required for critical events in meiosis and spermiogenesis.

Further, we have characterized global gene expression defects in TDP-43 cKO germ cells, shedding light on how TDP-43 cKO leads to male infertility[6,7]. We note that whereas TDP-43 and NELF-B cKO animals exhibit similar histological defects and failure to complete meiosis, more genes are affected by loss of TDP-43, likely due to its pleiotropic roles in transcription regulation, RNA processing and stability. While of high interest for its protein aggregation in diseases such as amyotrophic lateral sclerosis[46], this study provides new insights into the normal function of TDP-43 as it regulates gene expression through its RNA-binding domains[47–50].

Interestingly, loss of either NELF-B or TDP-43 causes significant decreases in expression of *Spo11* and other factors involved in meiosis. The SPO11 protein is critical for initiation of DSBs during meiosis, and accordingly, *Spo11* mutant mice fail to progress through meiosis, leading to infertility[12]. Further, a number of genes involved in DSB processing and repair are not fully induced in NELF-B cKO SC, and we find evidence of inefficient DSB repair in NELF-B cKO animals (Fig. 8). Our data suggest that NELF is required for appropriate expression of

genes needed for progression of SC through meiosis. Given our findings that DSBs generated by SPO11 correlate with promoter-proximally paused Pol II in SG, we propose that the loss of pausing in NELF-B cKO SG causes defects in SPO11-mediated DSB formation, impacting meiotic progression and timely DSB repair. In the future, it will be exciting to use these new data sets to expand our knowledge of key regulators of spermatogenesis, and to leverage these new insights into the roles of pausing in germ cell development towards novel therapeutic approaches to treat male infertility.

## Methods
Mice used in our research received humane care and were maintained at the College of Veterinary Medicine, University of Illinois Urbana Champaign. Experimental procedures involving mice were conducted per ethical guidelines listed in the study protocol and approved by the Institutional Animal Care and Use Committee (IACUC) of the University of Illinois Urbana Champaign.

### Statistics & reproducibility
In determining sample sizes, we made all possible efforts to minimize animal suffering. To isolate spermatogonia, spermatocytes, and round spermatids for PRO-seq and bulk RNA-seq experiments the number of

wild type male C57BL/6 J mice was determined based on the number of germ cells required for PRO-seq and RNA-seq. Where conditional knockout (cKO) mice were used for single cell RNA-seq we were mindful of the cost, time, and difficulty obtaining mutant mice of the required genotype. For the single cell RNA-seq, histology, and immunohistochemistry, and immunofluorescence experiments, $n = 3$ was used for experimental and control mice except as noted. The results were similar between biological replicates. No statistical method was used to predetermine sample size. No data were excluded from the analyses. The experiments were not randomized. The Investigators were not blinded to allocation during experiments and outcome assessment.

The statistical test used, error bars, and n values are defined in each figure legend. For box plots, line represents median, box represents 25–75th percentile, whiskers represent 1.5X interquartile range. P-values were calculated in R. For Venn diagrams, exact P-value for the overlap of gene lists was calculated using the hypergeometric distribution with the phyper function in R. For image quantification, P-values are from unpaired t-test, two-tailed calculated in GraphPad Prism.

## Mouse strains

C57BL/6 J (wild type control), and *Stra8-iCre* [Tg(*Stra8-icre*)1Reb/J, RRID:IMSR_JAX:008208] mice were obtained from Jackson Laboratory (Bar Harbor, ME, USA). Floxed *Tardbp* mice (mix of C57BL/6 N and C57BL/6 J) in which the third exon of *Tardbp* was flanked by loxP sites were obtained from Dr. Philip C. Wong of Hopkins University. Floxed *Nelfb* mice (C57BL/6 N) in which exons 1–4 were flanked by loxP sites were reported previously[36]. All mice used in this study were backcrossed to C57BL/6 J (B6) background for eight generations. *Tardbp* (TDP-43) and *Nelfb* (NELF-B) conditional knockout (cKO) mice were generated on pure B6 background. Crosses were set up to obtain pups with F/-, iCre genotype (one floxed *Tardbp* or *Nelfb* and one null allele, and transgenic for *Stra8-iCre*). Male pups with F/-, iCre genotype (referred to as cKO) would have the gene deletion in SG starting at PND4. For RNA-seq and PRO-seq experiments, spermatogonia, spermatocytes, and round spermatids were isolated from the wild type C57BL/6 J male mice at the ages indicated in the Results section of the manuscript. For single cell RNA-seq experiments control and cKO mice at postnatal day 24 (PND24) were used. Since the subject of this study was male germ cell differentiation and spermatogenesis; only male mice were used. Mice were maintained in breeding cages at a constant temperature of 21 °C with 50–60% humidity under a 12 h dark and 12 h light cycle. Food and water were made available ad libitum.

## Genotyping

Primers used for genotyping TDP-43 and NELF-B cKO mice (flox, null, iCre):

Oligo Name Sequence (5'-3')
*Nelfb* Flox-Fw TTTCCATCCTCCCCAGACACG
*Nelfb* Flox-Rv CAAACTCAGACCCTCTGCTTCC
*Nelfb* Null-Fw TTTCCATCCTCCCCAGACACG
*Nelfb* Null-Rv GCCAGAGGTGGTGTGTATGC
*Tardbp* Flox-Fw AACTTCAAGATCTGACACCCTCCCC
*Tardbp* Flox-Rv GGCCCTGGCTCATCAAGAACTG
*Tardbp* Null-Fw TCTTACAATGCCTGGCGTGGTG
*Tardbp* Null-Rv CGTGGTTGCGCACCCTAACTATAA
*iCre* Fw GCTCCTGTCTGTGTGCAGAT
*iCre* Rv CATCACCAGGGACACAGCAT

## Histology and immunohistochemistry

For hematoxylin−eosin (H&E) staining, mouse testes were fixed in Bouin's solution (Sigma, MO) overnight at room temperature (RT). The fixed tissues were washed in ice-cold 70% ethanol and paraffin-embedded using a Tissue-Tek VIP 1000 processor (Sakura Finetek,

Torrance, CA), sectioned at 4 μm with a Leica RM2125 RTS rotary microtome (Leica Biosystems, Buffalo Grove, IL) and mounted on glass slides. These were deparaffinized with xylene and hydrated through a series of graded ethyl alcohols. For morphological analysis slides were stained with Hematoxylin and Eosin. For immunohistochemistry with NELF-B and SP-10 antibodies, the testes were fixed overnight in PFA (Paraformaldehyde in 4% PBS, Thermo Scientific, Waltham, MA) and Bouin's solution (Sigma-Aldrich, St. Luois, MO), respectively and processed as above. Antigen retrieval was done using citrate buffer pH 6.0 in a vegetable steamer for 60 min. TBS-tween was used as the buffer rinse throughout the staining procedure. Endogenous peroxidase was blocked using 3.0% hydrogen peroxide for 10 min. Nonspecific background blocking was performed using Background Punisher (Biocare Medical, Pacheco, CA). The sections were incubated for 1 h at room temperature in Rabbit polyclonal anti-NELF-B rabbit polyclonal antibody (Proteintech (Rosemont, IL, catalog number 16418-1-AP)) at 1:100 dilution (primary antibody) or SP-10 guinea pig polyclonal antibody (In-house[51]) at a dilution of 1:1000. The primary antibody was omitted from one section each for a negative control. Following rinsing, the sections were incubated in HRP-conjugated anti-Rabbit secondary antibody (Jackson Immunoresearch Laboratories, PA Code Number: 111-035-144) at a 1:200 dilution for NELF-B staining, or Peroxidase-conjugated AffiniPure Donkey Anti-Guinea Pig IgG (H + L) (Jackson ImmunoResearch Laboratories, Code Number: 706-035-148) at a 1:200 dilution for SP-10 staining. DAB (3,30Diamenobenzidine) (Innovex Biosciences Inc., Richmond, CA) was used as the chromogen with an incubation time of 5 min. Slides were counterstained with hematoxylin, dehydrated, cleared and mounted.

## Isolation of mouse spermatogonia

Testes were collected from 25 male pups (6–8 days old) in 60 mm Petri dishes with ice-cold DMEM. Testes were decapsulated and minced randomly into 1–3 mm pieces. Pieces were transferred to a 15 ml tube with 10 ml of DMEM and centrifuged for 1 min at 100 x g. After an additional wash with DMEM, the pellet was left with 2.5 mL DMEM and 500 μl of Collagenase 1%, and 10 μl of 1% DNase added to the tube. The solution was transferred to a 25 ml flask and incubated at 34.5 °C with shaking (80 rpm) until tubules were well separated (~20 min). The enzymatic reaction was blocked with cold DMEM and centrifuged for 1 min at 100 x g twice. The pellet was left with 2 ml of DMEM and 2 ml of trypsin 0.25%, 350 μl of 1% collagenase, 750 μl of 1% hyaluronidase, and 10 μL of 1% DNase added to the tube. After incubation and washing as above, the enzymatic reaction was blocked with cold DMEM and filtered through a 100 μm mesh and spun at 500 x g for 5 min. The pellet was resuspended in 10 ml of DMEM, 10% SFB, 1% Glutamine, and 0.5% antibiotics, and cells were plated in a Matrigel-coated flask at 33.5 °C to allow Sertoli cells to attach. After 1 h, the culture supernatant containing the spermatogonia was centrifuged for 5 min at 500 x g and the pellet was washed and resuspended in 1 ml of PBS for cell counting. This yielded the spermatogonia cells.

## Isolation of mouse spermatocytes and round spermatids

Spermatocytes and round spermatids were isolated by the density gradient method known as STA-PUT as previously described[7]. Briefly, testes from 10 three-month-old C57BL/6 J mice were decapsulated using forceps, and tubules were collected in a 10 cm dish and washed in 10 ml of DMEM. Tubules were dissociated in 10 mg of collagenase and 20 μg of DNase in 8.5 ml of DMEM for 10 min in a 37 °C incubator with gentle agitation. Tubules were washed twice with cold DMEM. Germ cells were released by enzymatic treatment with 7 mg of collagenase, 15 mg of hyaluronidase, 10 mg of trypsin, and 20 μg of DNase in 8.5 mg of DMEM for 10 min in the 37 °C incubator with gentle agitation. The solution was transferred to a 50 ml conical tube, reconstituted in 45 ml of DMEM, and allowed to sediment for 10 min on ice to separate the heavier tubule pieces away from the germ cells. The

supernatant containing germ cells was transferred to a fresh conical tube and centrifuged at $900 \times g$ for 10 min at 4 °C. The cells were washed twice with PBS and loaded onto a 2–4% BSA density gradient to separate the larger spermatocytes and smaller round spermatids by gravity sedimentation for 3 h at 4 °C. Fractions (300 drops per fraction) of the heavier spermatocytes first followed by lighter round spermatids were collected over a 1 h period. Every fifth fraction of approximately 70 total fractions was observed under the light microscope to identify cells as spermatocytes or round-spermatids based on morphology. Fractions containing 90–95% pure populations of spermatocytes or round spermatids were pooled separately and centrifuged at 900 x g for 20 min at 4 °C and the pellets were processed as needed for RNA-seq and PRO-seq.

## Cell Permeabilization
For PRO-seq, cells were permeabilized. Cells ($2 \times 10^7$) were gently suspended in 1 ml of washing buffer (10 mM Tris-Cl pH 8.0, 10 mM KCl, 250 mM sucrose, 5 mM MgCl2, 0.5 mM DTT, 10% glycerol) and filtered through a 40 μm mesh. The mesh was washed with 9 ml of permeabilization buffer (washing buffer + 0.1% Igepal CA-360) to recover the remaining cells, and the 10 ml cell solution was incubated for 1 min at 25 °C while gently inverting the tubes. Then, the cell solution was centrifuged for 5 min at $500 \times g$ at 25 °C. The supernatant was discarded and the cell pellet was suspended in 1 ml of freezing buffer (50 mM Tris-Cl pH 8.0, 5 mM MgCl$_2$, 0.05 mM DTT, 40% glycerol). Cells were counted and permeabilization was ensured by Trypan blue staining. Then, cell solutions were centrifuged, and the pellet was suspended such that 100 μl contained 1 million permeabilized cells. Aliquots with 5 million cells each were snap-frozen in liquid nitrogen and stored at −80 °C for downstream analysis.

## PRO-seq library construction
Aliquots of frozen (−80 °C) permeabilized cells were thawed on ice and pipetted gently to fully resuspend. Aliquots were removed and permeabilized cells were counted using a Luna II, Logos Biosystems instrument. For SG, 1 million permeabilized cells were used for nuclear run-on, with 50,000 permeabilized *Drosophila* S2 cells added to each sample for normalization. For SC and RS, 2 million permeabilized cells with 100,000 S2 cells were used. Nuclear run-on assays and library preparation were performed essentially as described in previously[52,53]. Run-on reactions were performed at 30 °C in 2X nuclear run-on buffer (10 mM Tris (pH 8), 10 mM MgCl2, 1 mM DTT, 300 mM KCl, 20uM/ea biotin-11-NTPs (Perkin Elmer), 0.8 U/uL SUPERase-In (Thermo), 1% sarkosyl). RNA was then isolated using the Total RNA Purification kit (Norgen Biotek) according to manufacturer's protocol. Chemical fragmentation, adaptor ligations, and reverse transcription were performed as previously[53]. Eluted cDNA was amplified 5-cycles (NEBNext Ultra II Q5 master mix (NEB) with Illumina TruSeq PCR primers RP-1 and RPI-X) following the manufacturer's suggested cycling protocol for library construction. A portion of pre-CR was serially diluted and for test amplification to determine optimal amplification of final libraries. Pooled Libraries were sequenced at The Bauer Core Facility at Harvard University on an Illumina NovaSeq 6000 using an S1 flow cell and a paired-end 50 bp run.

## Bulk RNA-seq sample and library preparation
Isolated SG, SC, and RS cells were counted, and $1 \times 10^6$ cells were resuspended in 1 mL of Trizol and spiked with 1 uL of 1:10 diluted ERCC Spike-in Mix (Invitrogen). RNA was purified according to manufacturer protocol and RNA integrity was confirmed using a TapeStation (Agilent). 210 ng of RNA from each sample was diluted in 10 uL of water as input RNA for library generation using the TruSeq Stranded Total RNA sequencing kit with RiboZero rRNA depletion (Illumina). Manufacturer instructions were followed except for substitution of SuperScriptIII for SuperScriptII. The final libraries were amplified to 10–12 cycles and

purified using AMPure XP beads (Beckman Coulter). Pooled libraries (10 nM) were sequenced (paired-end 150 cycles) on an Illumina HiSeq 4000 at Novogene.

## Preparation of single cell suspensions from mouse testis
PND 24 (post-natal day 24) mice of TDP-43 cKO ($N = 3$), NELF-B cKO ($N = 2$) and corresponding littermate/ non-littermate control ($n = 4$) were used for single-cell RNA-seq. Testes were dissected and collected in 35 mm plate with ice cold DMEM (DMEM henceforth). Testes were decapsulated and the seminiferous tubules were minced thoroughly with micro squeeze scissors. The resulting minced tubules were collected in a 15 ml conical centrifuge tube with 10 ml DMEM and washed twice at $100 \times g$ for 1 min in an Eppendorf swing bucket centrifuge (model 5702 R). After the final wash the pellet was suspended in 1 ml DMEM and digested with 250 μl of 1% collagenase and 10 μl of 1% DNase at 35.5 °C for 20 min in a rotating shaker (100 rpm) until tubules were well separated. The enzymatic reactions were stopped by diluting the enzymes with excess amount of (12 ml) DMEM and washed twice at 100 x g for 1 min. After the final wash the pellet was suspended in 1 ml DMEM and put through a second enzymatic digestion with 1 mL of 0.25% trypsin, 350 μl of 1% hyaluronidase, 200 μl of 1% collagenase, and 10 μl of 1% DNase, at 35.5 °C for 20 min in the rotating shaker. The enzymatic digestions were stopped by adding 12 ml of DMEM. The resulting single cell suspensions were filtered through 40 μm EZFlow nylon mesh (Foxx Life Sciences, Londonderry, NH) and the filtered samples were centrifuged at 300 x g for 5 min. The supernatant was discarded carefully, and single cells pellet was suspended in DMEM with 1-2% BSA. Finally, the cell suspensions were passed through Flowmi 40 μm cell strainer (SP Bel-Art, Wayne, NJ) to discard clumped cells and collected in 1.5 ml microfuge tubes. The tubes were stored on ice and proceeded for the 10X genomic facilities.

## Construction of 10 × 3' RNA Single Cell libraries
Single-cell 3' cDNA libraries were prepared at the DNA Services laboratory of the Roy J. Carver Biotechnology Center at the University of Illinois at Urbana-Champaign. Single-cell suspensions were delivered to the facility and were counted and checked for viability > 80% using the Nexcelom K2 brightfield/dual florescence cell counter (Nexcelom Biosciences, Lawrence MA) with AO/PI staining, then washed with PBS buffer containing 2.0% BSA, then recounted for library preparation. The target number of cells (5000) from each population were converted into individually barcoded cDNA libraries with the Single-Cell 3' NextGEM v3.1 Chromium kit from 10X Genomics (Pleasanton, CA) following the manufacturer's protocols.

Following ds-cDNA synthesis, individually-barcoded dual-index libraries compatible with the Illumina chemistry were constructed. The final libraries were quantitated on Qubit (Life Technologies, Grand Island, NY) and the average size determined on the AATI Fragment Analyzer (Agilent Technologies, Santa Clara, CA). Libraries were pooled evenly and the final pool diluted to 5 nM final concentration. The 5 nM dilution was further quantitated by qPCR on a BioRad CFX Connect Real-Time System (Bio-Rad Laboratories, Inc. CA).

The final 10x single cell library pool was sequenced on the Illumina NovaSeq 6000 S4 flowcell as paired-reads with 150nt in length. The first read of the single-cell libraries is used for the UMI and 10x barcode only, the 2nd read contains the RNA sequencing information. Base-calling and demultiplexing of raw data was done with the mkfastq command of the software Cell Ranger 6.1.1 (10x Genomics). Sorted data was posted to a password-secured AWS site for download and downstream processing.

## Bulk RNA-seq mapping
Reads were quality filtered requiring a mean quality score >= 20 and trimmed to 100 nt. Reads were first mapped to the ERCC spike sequences using STAR 2.7.3a. Reads not mapping to spike were used

for alignment to mm10 using parameters --quantMode TranscriptomeSAM GeneCounts --outSAMtype BAM SortedByCoordinate --limitBAMsortRAM 42949672960 --outMultimapperOrder Random --outSAMattrIHstart 0 --outFilterType BySJout --outFilterMismatchNmax 4 --alignSJoverhangMin 8 --outSAMstrandField intronMotif --outFilterIntronMotifs RemoveNoncanonicalUnannotated --alignIntronMin 20 --alignIntronMax 1000000 --alignMatesGapMax 1000000 --outWigType bedGraph --outWigNorm None --outFilterScoreMinOverLread 0 --outFilterMatchNminOverLread 0. Duplicates were also removed using STAR.

Samples displayed variable recovery of spike-in reads between cell-types. Thus, spike normalization was used in place of DESeq2 size factors for each sample. These were calculated as mapped reference reads / mapped spike reads, and then a genome copy number correction was used for round spermatids to divide spike factors by 2. UCSC Genome Browser tracks representing read coverage were generated from the combined replicates in each condition after normalizing using the factors above.

## PRO-seq mapping

All custom scripts described herein are available on the AdelmanLab GitHub (https://github.com/AdelmanLab/NIH_scripts). Read pairs were trimmed using cutadapt 1.14 to remove adapter sequences (-O 1 --match-read-wildcards -m 20). An additional nucleotide was removed from the end of read 1 (R1), using seqtk trimfq (https://github.com/lh3/seqtk), to preserve a single mate orientation during alignment. The paired end reads were then mapped to a combined genome index, including both the spike (dm6) and primary (mm10) genomes, using bowtie2[54]. Properly paired reads were retained. These read pairs were then separated based on the genome (i.e. spike-in vs primary) to which they mapped. Reads mapping to the reference genome were separated according to whether they were R1 or R2, sorted via samtools 1.3.1 (-n), and subsequently converted to bedGraph format using a custom script (bowtie2stdBedGraph.pl). We note that this script counts each read once at the exact 3' end of the nascent RNA. Because R1 in PRO-seq reveals the position of the RNA 3' end, the "+" and "-" strands were swapped to generate bedGraphs representing 3' end positions at single nucleotide resolution. Agreement between replicates ($N = 3$ per condition) was determined by summing reads 150 bp downstream of TSSs and determining Spearman's correlation coefficients (Spermatogonia Spearman's rho > 0.96, Spermatocytes Spearman's rho > 0.95, and Round spermatids Spearman's rho >0.95).

Each sample bedGraph was normalized to the sample with the lowest spike reads with a genome copy number correction used for round spermatid normalization (spike factor / 2). Normalized replicates were then merged to generate bigwig files.

## Generation of transcript annotations

The Get Gene Annotation pipeline was used to generate high-confidence gene annotations based on PRO-seq and RNA-seq (https://github.com/AdelmanLab/GetGeneAnnotation_GGA; https://doi.org/10.5281/zenodo.5519927). A hybrid Ensembl/RefSeq GTF was used as a basis for gene annotations. Unnormalized reads from all nine PRO-seq samples ($n = 3$ for each cell type SG, SC, and RS) were used to refine TSS position for annotated TSSs based on 5' ends of PRO-seq data, and the bulk RNA-seq was used to define TESs. A minimum 5' PRO-seq read count of 8 and a search window of 1 kb was required for a gene to be considered active and for re-alignment of the annotated TSS to the position with maximal nascent RNA 5' end reads. This generated a list of $n = 20,015$ active promoters from our pipeline, including 16,376 protein coding genes, 3145 long noncoding RNAs (lncRNA), and the rest comprised of other small RNA's and biotypes.

## scRNA-seq analysis

Reads were mapped with 10x Genomics Cell Ranger 7.0.0 to the 2020-A version of the mm10 pre-built reference[55]. Seurat v4.2.0 was used for downstream processing[56], which mostly followed a tutorial offered by the Harvard Chan Bioinformatics Core (https://doi.org/10.5281/zenodo.5826256). This included filtering to keep cells with more than 1000 detected genes and less than 10% mitochondrial transcripts. A gene needed to be detected in more than 15 of these retained cells, resulting in 45,132 cells and 26,567 genes being considered for downstream steps.

Cells were integrated across conditions using SCTransform normalization, in which no additional variables were regressed out. The top 40 principal components were used for cell clustering which had a selected resolution parameter of 0.8. This step yielded 30 clusters which were subsequently identified using marker genes from two earlier studies of mouse spermatogenesis[11,39]. The FindConservedMarkers() function in Seurat helped assign cell types to some clusters that could not be obviously defined from previously highlighted marker genes. We removed one cluster that did not have any reliable conserved markers. Additionally we removed 6 other clusters because of poor quality, resulting in a total of 23 clusters representing 27,578 cells. Further, we removed germ cells expressing NELF-B and TDP-43 from the NELF-B cKO, and TDP-43 cKO samples, respectively. This resulted in 23 clusters representing 24,269 cells which were consolidated into 10 cell type classifications used by another study[11]. Additionally, we also identified the sub stages of spermatocytes (preleptotene:pL, leptotene:L, zygotene:Z, pachytene:P, and diplotene:D) within the SC cluster using markers from earlier studies[38,39]. Cell numbers per cell type were normalized by the number of mice per condition to allow direct comparisons across genotypes. The number of cells per genotype are Control $n = 11,170$, NELF-B cKO $n = 6,404$, and TDP-43 $n = 6,695$.

Differentially expressed genes in SG and early meiotic cells (pL, L, and Z) across conditions were determined via a pseudobulk analysis, using the pairwise Wald test within DESeq2 v1.36.0[57]. Counts were first aggregated across SG, and early meiotic cells to collapse resolution to the level of biological replicates (Control: $n = 4$, NELF-B cKO: $n = 2$, TDP-43 cKO: $n = 3$), and the corresponding sample-level matrix was provided as input to DESeq2. Significance was assessed with a padj < 0.01 threshold and the absolute value of shrunken log2 fold changes was required to be > = 1.

Over/underrepresentation of scRNA-seq DE genes across the bulk clusters was first calculated as the proportion of DE genes within each bulk cluster. These values were then normalized by the relative sizes of each bulk cluster, permitting direct comparisons between the clusters.

## Differentially expressed genes in RNA-seq

Read counts were calculated on a per-gene basis in a strand-specific manner using featureCounts in R version 3.6.1[58]. Differentially expressed genes were identified with DESeq2 using coverage counts from above. For comparisons between cell types (SG versus SC and SC versus RS), an adjusted $p$-value threshold of 0.05 and fold change > 1.5 was used.

## Differentially expressed genes in PRO-seq

Read counts were calculated per gene, in a strand-specific manner, based on the annotations described above, using the custom script make_heatmap (available at https://github.com/AdelmanLab/NIH_scripts; https://doi.org/10.5281/zenodo.5519914). This quantification procedure includes signal from the dominant TSS to TES. Differentially expressed genes were identified using DESeq2. Spike normalization factors were enforced. An adjusted $p$-value threshold of < 0.05 and fold change > 1.5 was used as in the RNA-seq analysis.

## Cluster analysis

All genes that showed differential expression in any cell type were evaluated ($n = 17,141$). The relative expression of each gene in each cell type was calculated as a fraction of the cell type with the maximum RNA-seq levels, where the highest cell type = 1. The genes were clustered using K-means and standard Euclidean distances into 6 clusters. PRO-seq relative gene body (TSS + 250 to TES) and promoter (TSS to TSS + 150) window signals were calculated similarly. Genes that were p-adj. = "NA" in the PRO-seq DESeq2 output were filtered out to generate a final differentially expressed gene list of $n = 17,078$ genes used in downstream analysis.

## Functional enrichment of gene categories

Gene sets were queried for enriched gene ontology categories using clusterProfiler v4.4.4[59]. Considered categories were from the Biological Process (BP) subontology and p-values were corrected via the Benjamini & Hochberg method (BH). Significant categories satisfied a qvalueCutoff = 0.05 and were subsequently consolidated on the Revigo website[60], with representative categories reported.

## Motif analysis

Sequences of promoter regions 100 nt upstream of the TSS were obtained using our GGA coordinates and the UCSC table browser tool in the mm10 genome. The TATA motif position weight matrix was from ElemeNT[61], and was run in the Find Individual Motif Occurrences (FIMO) tool in the MEME suite using default options[62]. Results filtered to a p-value ≤ 0.005 and number of genes with matches was used to calculate percent.

## Double-strand break analysis

SPO11-oligo raw total mapped reads in SC from an earlier study were merged across the B6 and Atm wt genotypes[40], and quantified over active TSSs using make_heatmap, as described above. To compare these read distributions across SG promoter count quartiles, promoters with < 5 reads in SG were filtered out ($n = 14,803$ remaining) prior to defining quartiles. SPO11 signal was summed over the +/−500nt window relative to TSSs. H3K4me3 ChIP-seq reads in B6[44], re-used by that same study[40], were also summed across the +/−500nt window relative to active TSSs.

## Metagene analysis

Composite metagene distributions were generated by summing sequencing reads within bins at each indicated position with respect to the TSS and dividing by the number of TSSs included within each group. For PRO-seq, bins are 20 nt. For SPO11 and H3K4me3, bin size is 50 bp. These were plotted around the TSS at distances indicated in figure legends. Heatmaps of relative RNA-seq and PRO-seq signals, SG PRO-seq, and SPO11 oligo data were generated using Partek Genomics Suite version 7.19.1125.

## Preparation of chromosomal spreads for meiotic spermatocytes

Chromosome spread slides were prepared as described previously[63]. Briefly, testes were dissected and submerged in a phosphate-buffered saline (PBS) solution. The seminiferous tubules were gently squeezed out from the tunica albuginea, minced in 50 μl of PBS on a depression slide until a cloudy cell suspension is formed. Next, the cell suspension was transferred to a 1.5 ml Eppendorf tube containing 1 ml of PBS. After centrifugation, the supernatant was carefully aspirated and 80 μl of 0.1 M sucrose was added to resuspend the testicular cells.

Prior to spreading these testicular nuclei, glass slides were coated with a solution comprising 1% paraformaldehyde (PFA) and 0.1% Triton X-100. On each slide, 18 μl of sucrose cell suspension was gently added to the PFA solution, ensuring an even distribution across the slide surface. Subsequently, the slides were placed in a sealed humid chamber for an overnight incubation. The next morning, the chamber was kept ajar for 30 min, after which the slides were taken out of the chamber to facilitate thorough drying. Thereafter, the slides were immersed in a Coplin jar filled with distilled water, undergoing a 5 min agitation on a shaker at room temperature. After undergoing two additional washes in a 0.4% Photo-Flo 200 (Kodak; 1464510) solution, the slides were taken out and left to air dry. Finally, these slides were either immuno-stained immediately or stored in a −80 °C freezer for future use.

## Immunostaining of meiotic chromosomal spreads

Meiotic chromosomal spreads obtained from both NELF-B cKO and WT mice were subjected to immunostaining per ref. 64. Briefly, the chromosomal spreads underwent two washes with Tris-buffered saline containing 0.1% Tween-20 (TBST). Next, the slides were subjected to two 15 min incubations with 250 μl of 10% antibody dilution buffer (ADB) blocking solution (0.3% bovine serum albumin, 10% normal goat serum, and 0.005% Triton- X-100 in TBS). Each slide was further applied with 100 μl of primary antibodies: anti-SYCP3 (Abcam; 15093) diluted at 1:200; anti- γH2AX (Millipore Sigma; 05-636) diluted at 1:500; or anti RPA (Abcam; 76420) 1:200. After exposure to antibodies, the slides were covered with plastic cover slips and incubated overnight in a dark, humid container at room temperature. Coverslips were detached by gentle peeling using tweezers. Subsequently, slides were incubated in 10% ADB, repeated twice for 15 min each. Goat secondary antibodies (anti-mouse 594 [Molecular Probes; A11020; 1:1000 dilution]; anti-rabbit 488 [Molecular Probes; A11070; 1:1000 dilution]) were diluted in ADB and applied onto the slides. These slides were then covered with plastic coverslips and incubated in a dark, moist box at 37 °C for 1 h. Afterward, the slides were taken out of the incubator and allowed to equilibrate to room temperature. The coverslips were carefully removed, followed by three washes with TBST. After completing the TBST washes, the slides underwent two additional washes with distilled water and left to air dry for 5 min. Finally, slides were mounted with 25 μl of Prolong mounting media (Fisher Scientific; P36970) containing 4′,6-Diamidine-2′-phenylindole dihydrochloride (DAPI; Millipore Sigma; D9542), and then covered with a glass coverslip.

## Imaging and data analysis

Spermatocyte nuclei were captured using Keyence BZ-X microscope. Subsequently, the acquired images were subjected to quantitative analysis using ImageJ software. The areas of γ-H2AX and DAPI were measured using ImageJ before calculating the ratio of γ-H2AX to DAPI areas. ImageJ was also applied to count the RPA foci.

## Reporting summary

Further information on research design is available in the Nature Portfolio Reporting Summary linked to this article.

# Data availability

Bulk RNA-seq, PRO-seq, and scRNA-seq data generated in this study were deposited in the NCBI GEO database under accession code GSE228454. Previously published SPO11 oligo data is available at the NCBI GEO database under accession code GSE84689[40]. Published H3K4me3 ChIP-seq data is available at the NCBI GEO database under accession code GSE52628[44]. The mouse reference genome mm10 is publicly available from UCSC https://hgdownload.soe.ucsc.edu/goldenPath/mm10/bigZips/. Source data are provided with this paper.

# Code availability

No new scripts were generated for this study. Custom scripts including make_heatmap, bowtie2stdBedGraph.pl, and normalize_bedGraph are all publicly available (https://github.com/AdelmanLab/NIH_scripts; https://doi.org/10.5281/zenodo.5519914). The Get Gene Annotation script is publicly available (https://github.

com/AdelmanLab/GetGeneAnnotation_GGA; https://doi.org/10.52
81/zenodo.5519927). Additionally, the scRNA-seq tutorial from the
Harvard Chan Bioinformatics Core is publicly available (https://doi.
org/10.5281/zenodo.5826256).

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

## Acknowledgements

The authors would like to thank the Nascent Transcriptomics Core at Harvard Medical School, Boston, MA for performing PRO-seq library construction. We also thank the HMS Biopolymers Facility, Bauer Core Facility at Harvard University and Novogene for sequencing. We are grateful to the DNA Services laboratory of the Roy J. Carver Biotechnology Center at the University of Illinois at Urbana-Champaign (UIUC) for scRNA-seq sample prep and sequencing. We acknowledge the services of the Histology Core of the College of Veterinary Medicine, UIUC. This work was supported by the National Institutes of Health (NIH R01HD36239 to P.P.R., R01GM135549 to H.Q., and NIH R01HD094546 to P.P.R. and K.A.)

## Author contributions

Mouse generation and male germ cell purifications H.D.Z., RNA-seq library generation, analysis, and visualization, E.G.K., PRO-seq analysis and visualization, E.G.K., scRNA-seq sample generation, D.R., scRNA-seq data analysis and visualization K.B. and G.M.N., Immunohistochemistry and histology, P.P.R. and I.I.J., Immunofluorescence of meiotic spreads R.R.T. and H.Q. Funding acquisition and supervision, K.A. and P.P.R.

## Competing interests

K.A. received research funding from Novartis not related to this work, is a consultant for Odyssey Therapeutics, and is on the SAB of CAMP4 Therapeutics. The remaining authors declare no competing interests.
