## [Peer Review File · Nature Communications]

RNA polymerase II pausing is essential during spermatogenesis for appropriate gene expression and completion of meiosisREVIEWER COMMENTS

Reviewer #1 (Remarks to the Author):

In this study Kaye et al perform an in-depth analysis of transcriptional dynamics during mouse spermatogenesis, integrating bulk RNA-seq, PRO-seq, and single cell RNA-seq data with analysis of transcriptomic and phenotypic changes in NELF-B and TDP-43 germline mutants. They reach several novel conclusions, including the finding that RNA Polymerase II promoter-proximal pausing is required at the spermatogonial stage to establish appropriate expression profiles for sperm differentiation genes, that loss of NELF-B and TDP-43 induce arrest at the pachytene stage and complete loss of germ cells over time, and (surprisingly) that paused Pol II is associated with binding of SPO11 and double strand break (DSB) formation. On balance the study is a valuable step forward in understanding the complex transcriptional dynamics of mammalian spermatogenesis. There are some cases where interpretations are not strongly supported by the data or potential artifacts must be addressed, including discrepancies between the NELF-B and TDP-43 phenotypes, normalization of RNA-seq data, some gaps in description of the scRNA-seq analysis, and the nature of the relationship between SPO11-induced DSBs and paused Pol II.

Major comments:

- 1) The use of spike-ins to normalize the RNA-seq data is valuable and an improvement over most published spermatogenesis RNA-seq datasets. Similarly, differences in chromosome content across spermatogenesis is an important consideration for normalization that is taken into account. However, while it is true that round spermatids have 1C and spermatogonia have 2C chromosome content as stated, spermatocytes sorted by STA-PUT have 4C content, since they have undergone premeiotic DNA replication but neither the first nor second meiotic divisions. This difference between spermatogonia and spermatocytes should also be accounted for during normalization.
- 2) It is not clear what analysis of the TDP-43 mutant phenotype adds to this study. TDP-43 has been suggested to regulate promoter-proximal pausing of Pol II but its role in this process is not as clear-cut as NELF-B. The transcriptional profile and mutant phenotype shown in this paper are similar in some ways to that of NELF-B, but not the same. Throughout the text they are discussed as if they reinforce each other. Better attention should be paid to the actual phenotype of TDP-43 and ways in which it is both similar to and different from the NELF-B cKO. For example, in Figure 5b, there is a more severe depletion of spermatocytes in the TDP-43 compared to the NELF-B mutant. There are substantial differences in the sets of differentially expressed genes described in Fig 6, although both NELF-B and TDP-43 appear to regulate SPO11. The model that TDP-43 may be a stage-specific pause release factor as suggested in lines 436-441 is not very well supported by the data as shown. On the other hand, the NELF-B mutant phenotype supports most of the central conclusions of the study.
- 3) Line 141-142: "we hypothesize that genes expressed late in spermatogenesis may be preferentially short, to enable rapid RNA synthesis and transcript accumulation prior to spermiogenesis". In order to make this statement, other confounders should be excluded or at least discussed. For example, if many members of a large gene family are coordinately regulated at a given stage, their lengths might dominate this analysis independently of any selection for faster RNA synthesis.
- 4) The complete depletion of germ cells in NELF-B cKO testes at days PND35 and adult stages implies that there is a spermatogonial stem cell defect in addition to the meiotic arrest discussed in the paper. This should be acknowledged and discussed.
- 5) The observation that paused Pol II is associated with sites of SPO11-mediated DSBs is potentially exciting. However, this association could be explained by independent correlation of SPO11 and paused PolII with open chromatin or other factors, rather than any mechanistic relationship between them. The correlation alone does not robustly support the conclusion that "loss of pausing in NELF-B

cKO SG causes a collapse of promoter chromatin architecture, causing defects in SPO11-mediated DSB formation" (lines 455-457).

6) Methods relating to the 10x scRNA-seq data are relatively sparse. In addition, it is surprising that there are so few round spermatids in the control. This could be because of the juvenile stage of the testes or because of clustering artifacts. UMAP projections and/or violin plots of the markers used for cluster assignment should be shown in the supplement so the accuracy of cluster assignments can be more easily evaluated. The number of replicates should be specified and also how replicates (if any) were treated in downstream analysis.

Minor comments:

7) "Mature spermatids" (line 24) is confusing; sperm are considered mature while spermatids are still maturing.

8) It would be helpful for the authors to be more explicit about the assumptions regarding when in spermatogenic development pause release is thought to occur. Spermatocytes remain in meiotic prophase for about two weeks, and the STA-PUT sorted fraction includes cells from across this time frame. Transcriptional activation by pause release is thought to be important for rapid activation. In the scenario proposed by the authors, would pause release occur at meiotic entry as implied by its importance for SPO11 activation? If so, this could be more explicitly described in the discussion.

9) Some quantification should be provided for the Spo11 PRO-seq browser tracks shown in Figure 6i; as shown, it is hard to distinguish differences between the tracks from background noise.

10) Figure panels 1k and S1d appear to be the same. If there is a difference, this should be better explained in the figure panel and legend.

11) In Figure 3, it is claimed that there is similar PRO-seq signal at promoter and gene body in clusters 1 and 2. This looks correct for cluster 1, but in cluster 2 there are quite a few genes with high signal at both gene body and promoter in SC and only at the promoter in SG.

12) In Figure 4b, there are some positively-staining cells toward the lumen in the cKO image at PND15, which is not a typical location for Sertoli cells. Is this nonspecific staining, or are these germ cells that escaped Cre excision?

Reviewer #2 (Remarks to the Author):

This paper investigates the role of RNA polymerase II (Pol II) pausing during spermatogenesis and its impact on gene expression and meiotic processes. By combining bulk sequencing, PRO-seq, and single-cell RNA-seq techniques from wild-type (WT) mice and two conditional KO mouse strains: NELF or TDP, both known to affect Pol II pausing. The results highlight the essentiality of Pol II pausing in spermatogonia for the appropriate expression of genes crucial to spermatogenesis. Furthermore, the loss of NELF, an inducer of Pol II pausing, in immature germ cells leads to inhibited differentiation into round spermatids. Additionally, immunohistochemistry and scRNA-seq analysis in NELF and TDP43-deficient testes reveal an unexpected role of Pol II pausing in the regulation of meiosis and meiotic gene expression, including the double-strand break protein SPO11. This intricate interplay between RNA Pol II pausing and double-strand break formation adds a novel dimension to our understanding of spermatogenesis regulation. While further investigation is required to fully elucidate these mechanisms, this study presents intriguing preliminary findings that contribute significantly to the germ cell research community.

COMMENTS:

1. The observation that spermatid genes do not accumulate paused Pol II, even in spermatogonia or spermatocytes, is intriguing. Further investigation is needed to determine the differences in RNA Pol II complex recruitment between round spermatid genes and earlier germ cell genes. It would be interesting to explore whether these differences are related to TBP-dependent or TBP-independent gene regulation.
2. It would be beneficial to investigate the expression patterns of NELFb and TDP43 proteins/RNA to gain a better understanding of their roles in spermatogenesis. Examining their expression profiles in different germ cell populations and at various stages of development could provide valuable insights into their functions.
3. In the absence of NELF or TDP43 mutants, it would be intriguing to determine if the paused spermatogonia (SPG) genes identified by bulk PRO-seq in wild-type samples are prematurely expressed or if the loss of pause leads to gene expression loss. Additionally, it would be interesting to investigate whether the loss of NELF or TDP43 results in the accumulation of Pol II at a second pause site, as described in a previous study titled "NELF regulates a promoter-proximal step distinct from RNA Pol II pause-release." This finding would be particularly relevant in the context of double-strand break (DSB) formation.
4. The proportion of spermatocyte cells in NELF and TDP43 mutants appear to differ. It would be informative to examine whether these knockout (KO) strains exhibit different stages of arrest. Providing the fraction of Leptotene, Zygotene, and Pachytene spermatocytes would enhance the understanding of the specific developmental stages affected by the NELF and TDP43 mutations. Synchronization of spermatogenesis in mutants could potentially increase the number of stage specific germ cells numbers available for PRO-seq analysis.
5. Investigating whether the decrease in Spo11 levels in the NELF and TDP43 KO animals leads to changes in the number of DSBs during meiotic prophase in these mutants is crucial. As changes in SPO11 expression have been shown to modulate DSB numbers. Examining this relationship in the NELF and TDP43 mutants would provide valuable insights into the impact of Pol II pausing on DSB formation.
6. The correlation between SPO11-oligos and PRO-seq enrichment in promoter proximal regions in Spermatogonia is intriguing. Exploring the instructive nature of these PRO-seq peaks and investigating whether sites of DSB changes occur in the NELF and TDP43 KO strains where pausing is lost would be valuable. Additionally, determining where these germ cells arrest in meiotic prophase and assessing whether these mutants phenocopy the SPO11 knockout would further enhance our understanding of the relationship between Pol II pausing, DSBs, and meiotic progression.
7. Generating PRO-seq libraries from both wild-type and NELF mutants (and TDP43 mutants as well, although the number of Leptotene cells may be lower) would significantly strengthen the most exciting aspects of the paper. Synchronizing spermatogenesis stages could potentially help enrich the Leptotene/Zygotene stage, facilitating more comprehensive PRO-seq analysis and improving the robustness of the findings.

Reviewer #3 (Remarks to the Author):

Kaye et al., in this manuscript, described the roles of RNA polymerase II (Pol II) pausing in spermatogenesis and regulation of meiosis. They found thousands of differentially expressed genes during spermatogenesis. Interestingly, various clusters of genes are differentially expressed in

Spermatogonia (SG), Spermatocytes (SC), and spermatids (RS). PRO-seq analyses confirmed the transcriptional regulation of these clusters of genes during spermatogenesis. Accumulation of PRO-seq signals at promoter-proximal regions representing Pol II pausing correlated with gene activity during spermatogenesis. Furthermore, PRO-seq data showed an increased transcriptional initiation or faster release of paused Pol II complexes associated with overexpressed genes.

NELF-B conditional knockout (KO) mice showed vacuole formation and disorganization of meiotic cells as early as postnatal day 15 (PND15). Furthermore, the loss of germ cells and more vacuoles appeared at PND24. A further increase in germ-cell loss was correlated with aging in NELF-B KO mice. Single-cell RNA-sequencing (scRNA-seq) data showed a defect in germ cell maturation in NELF-B KO mice. This phenotype was similar to TDP-43 germ cell KO. Various genes were downregulated, including genes related to spermatid development and motility pathways, confirming the germ cell developmental defects in both NELFB and TDP-43 KO mice.

They further correlated the Pol II pausing (promoter activity) with sites of double-strand breaks. They found that NELF-B KO in SG both perturbs Spo11 gene activation as well as impairs the proper progression of meiosis.

This is a well-written manuscript. I have the following comments that could improve this manuscript further:

- 1) It's unclear if the gene size difference (Fig 1k) is statistically significant.
- 2) It's good to see RNA-seq and PRO-seq results corroborate; distinct mechanisms may be involved in gene regulation. It would be great to show a faster pause release of some representative genes (or genome-wide, if possible) using the Pol II decay curve after Triptolide treatments (time course).
- 3) How Pol II pausing is impacted after NELF KO in germ cells? Looking at the cell numbers in Fig 5c, it could be possible to do PRO-seq or CUT&RUN using SG and SC cells (cells could be pooled) after NELF KO and comparing them with the control samples. Results from these experiments could further strengthen the notion that, indeed, Pol II is essential during spermatogenesis for proper gene expression and completion of meiosis.
- 4) Data presented under the heading "Promoter activity in SG correlates with sites of double-strand breaks in SC" in the Result section is fascinating. Authors need to validate these findings by using their NELF KO models. Does disrupting Pol II pausing changes the sites of double-strand breaks?

We thank all three reviewers for their careful reading of our manuscript and the thoughtful and constructive suggestions made to improve this study. We have addressed all comments with new experiments and analyses wherever possible and have edited the text of our manuscript to clarify aspects that were sources of concern.

Our point-by-point responses to each reviewer comment are below. We hope that the reviewers agree that our manuscript is stronger as a result of these changes, and we appreciate their input.

Reviewer #1 (Remarks to the Author):

In this study Kaye et al perform an in-depth analysis of transcriptional dynamics during mouse spermatogenesis, integrating bulk RNA-seq, PRO-seq, and single cell RNA-seq data with analysis of transcriptomic and phenotypic changes in NELF-B and TDP-43 germline mutants. They reach several novel conclusions, including the finding that RNA Polymerase II promoter-proximal pausing is required at the spermatogonial stage to establish appropriate expression profiles for sperm differentiation genes, that loss of NELF-B and TDP-43 induce arrest at the pachytene stage and complete loss of germ cells over time, and (surprisingly) that paused Pol II is associated with binding of SPO11 and double strand break (DSB) formation. On balance the study is a valuable step forward in understanding the complex transcriptional dynamics of mammalian spermatogenesis. There are some cases where interpretations are not strongly supported by the data or potential artifacts must be addressed, including discrepancies between the NELF-B and TDP-43 phenotypes, normalization of RNA-seq data, some gaps in description of the scRNA-seq analysis, and the nature of the relationship between SPO11-induced DSBs and paused Pol II.

Major comments:

1) The use of spike-ins to normalize the RNA-seq data is valuable and an improvement over most published spermatogenesis RNA-seq datasets. Similarly, differences in chromosome content across spermatogenesis is an important consideration for normalization that is taken into account. However, while it is true that round spermatids have 1C and spermatogonia have 2C chromosome content as stated, spermatocytes sorted by STA-PUT have 4C content, since they have undergone premeiotic DNA replication but neither the first nor second meiotic divisions. This difference between spermatogonia and spermatocytes should also be accounted for during normalization.

We are glad that the reviewer agrees that proper normalization is important, and that spike-ins allow for more precise comparison of RNA levels between cell conditions compared to sequencing depth-based strategies. Typically, because the spike-ins are based on cell count, we use the spike factors as-is and this is sufficient to account for global differences in RNA/transcription levels. However, our initial quality control to assess the spike in normalization showed very few RS-specific genes, and key marker genes that should have highest expression in RS did not (see Reviewer Figure 1A, below). We thus investigated whether correcting for the haploid status of RS would better normalize the data and reflect the anticipated biology. Indeed, with this correction, we saw reasonably selective expression of RS marker genes in RS cells (Reviewer Figure 1B). We therefore implemented this correction for the haploid genome content of RS for both RNA-seq and PRO-seq.

Reviewer Figure 1. RNA-seq expression of four RS marker genes without (A, left) and with (B, right) correction for haploid genome content of RS. Correcting our RNA-seq data sets for the haploid (1N) status of RS yielded the anticipated enrichment of expression of RS marker genes in RS cells.

When receiving the reviews of our manuscript, we realized that we did not correct for the fact that STA-PUT enriches for SC that are 4C (largely pachytene SC). To follow the suggestion of the reviewer, we re-did our normalization and quality control evaluation steps, now introducing this additional normalization factor (that effectively divides the SC expression levels by 2). We note that this approach assumes that all SC isolated by STA-PUT are 4C and that all four copies of the genome are equally competent to be active.

Reviewer Figure 2

Reviewer Figure 2. RNA-seq expression of SC marker genes without (A, left) and with (B, right) correction for enrichment of SC cells with 4C genome content. Correcting our RNA-seq data sets for the 4C status of many SC isolated by STA-PUT yielded a loss of SC marker gene specificity.

Based on our evaluation, normalization that makes these assumptions creates some unanticipated consequences that we fear does not accurately reflect the underlying biology. 1) With this method, we observe a decrease of SC marker-gene specificity, using the top 10 SC marker genes from Reference #11 (Reviewer Figure 2, at left).

2) We clustered RNA-seq expression levels after dividing the SC signal in half to account for a 4C genome content. We maintain the previous 6 kinetic clusters, but with different distributions of genes per cluster (Reviewer Figure 3, at left). While the cell type-specific clusters are still present (clusters 1, 4, and 6), three of the SC marker genes from Reference #11 (Calm2, Lyar, cdc38) shifted from the SC-specific cluster 4 into cluster 5, which has intermediate SC and RS expression. Further, the relationship between cell types in expanded clusters 2 and 5 becomes confusing.

Reviewer Figure 3. Clustering of RNA-seq expression levels after accounting for 4C content of SC yields same kinetic profiles, but with unanticipated reduction in SC-specific gene expression. Clusters 3 and 4 were markedly reduced in size, with genes shifting into less cell-type specific clusters 2 and 5.

4) Finally, dividing SC PRO-seq signals by 2 to correct for 4C content causes nearly all promoters to exhibit a loss of PRO-seq signal in SC as compared to SG (Reviewer Figure 5, below). While we do believe that pausing is prevalent in SG and that pause release

3) Particularly odd is the effect of correction for 4C content in SC on Cluster 2 genes, which still show reduced expression by RNA-seq in RS as compared to SC, yet now have higher Gene body PRO-seq in RS as compared to SC (Reviewer Figure 4, below).

Reviewer Figure 4. A. Correction for 4C in SC causes broad reduction of SC-specific Gene body PRO-seq signal. Note the dearth of red color (signifying maximum signal) in SC cells in the new clusters following 4C correction. B. Also, note the profile of Cluster 2 Gene Body PRO-seq signal, which now appears to go back up in RS, despite decreased expression of these genes. This seems poorly reflective of the biology.

allows for gene activation in SC, we fear that normalizing for 4C content in SC over-exaggerates this effect to a level that does not reflect a biological reality.

Taken together, our analyses suggest that normalization of SC for 4C status causes artifacts that are not reflective of proper gene expression. We have thus opted to maintain our original normalization strategy using a combination of the spike-ins and a copy number correction for the haploid status of RS, but without assuming all four copies of chromosomes in pachytene SC cells are active and also require a correction factor. We hope the reviewer understands our rationale for this decision.

We emphasize that the key conclusions, such as the regulation of genes activated in SC by pausing, is maintained with either normalization strategy, and the apparent role of pausing in gene activation becomes even more enhanced with 4C correction. But, since other gene expression results are inconsistent with the anticipated biology when introducing an extra SC normalization factor, we have decided to keep the initial strategy of only correcting for the haploid RS levels.

Reviewer Figure 5. Correcting for 4C in SC exaggerates the enrichment of promoter-proximal pausing in SG, with nearly all genes losing promoter RNAPII signal in SC.

2) It is not clear what analysis of the TDP-43 mutant phenotype adds to this study. TDP-43 has been suggested to regulate promoter-proximal pausing of Pol II but its role in this process is not as clear-cut as NELF-B. The transcriptional profile and mutant phenotype shown in this paper are similar in some ways to that of NELF-B, but not the same. Throughout the text they are discussed as if they reinforce each other. Better attention should be paid to the actual phenotype of TDP-43 and ways in which it is both similar to and different from the NELF-B cKO. For example, in Figure 5b, there is a more severe depletion of spermatocytes in the TDP-43 compared to the NELF-B mutant. There are substantial differences in the sets of differentially expressed genes described in Fig 6, although both NELF-B and TDP-43 appear to regulate SPO11. The model that TDP-43 may be a stage-specific pause release factor as suggested in lines 436-441 is not very well supported by the data as shown. On the other hand, the NELF-B mutant phenotype supports most of the central conclusions of the study.

We agree that the data in this manuscript demonstrate that the mechanisms of action of TDP-43 and NELF are quite different. In the revised manuscript, we have worked to do a better job of comparing and contrasting the two phenotypes. In particular, we have clarified the language around TDP-43, removing the suggestion that it is a stage specific pause release factor, and highlighting the differences between the two KO animals. Specifically, we note that NELF is a pausing factor (as shown in many other studies) and the consequences of NELF-B loss are well in line with this role. In contrast, TDP-43 loss has more far reaching, likely pleiotropic effect on spermatogenesis, causing perturbations in gene expression across multiple stages of spermatogenesis and more severe depletion of spermatocytes.

3) Line 141-142: “we hypothesize that genes expressed late in spermatogenesis may be preferentially short, to enable rapid RNA synthesis and transcript accumulation prior to spermiogenesis”. In order to make this statement, other confounders should be excluded or at least discussed. For example, if many members of a large gene family are coordinately regulated at a given stage, their lengths might dominate this analysis independently of any selection for faster RNA synthesis.

To address this comment, and a related question from Reviewer #3, we have added p-values to the comparison of genes expressed late in spermatogenesis (cluster 6) versus other gene clusters, showing that cluster 6 genes are significantly shorter ($P < 0.001$).

In interpreting this result, we did consider confounders, in particular noting that many genes in cluster 6 are lncRNAs - which are often much shorter than mRNAs. We thus repeated the comparisons of gene length between clusters looking only at mRNAs (see Supplementary Fig 1d). This analysis showed that cluster 6 mRNAs are shorter than mRNAs in other clusters. We did not detect any other large gene family or RNA subtype that we could investigate further. This is now noted in the legend to Supplementary Fig 1d. We hope the reviewer agrees that, based on this data, raising as a hypothesis that genes expressed late in spermatogenesis may be preferentially short, is acceptable.

4) The complete depletion of germ cells in NELF-B cKO testes at days PND35 and adult stages implies that there is a spermatogonial stem cell defect in addition to the meiotic arrest discussed in the paper. This should be acknowledged and discussed.

Thank you for the comment. NELF-B is in fact highly expressed in spermatogonia. In response to another comment, we have now included a figure showing the expression pattern of NELF-B across spermatogenesis (Supplementary Fig. 5b). Stra8-iCre mediated excision of floxed genes starts in the undifferentiated spermatogonia on PND4 and reaches full penetrance by PND40 (Reference #37). Thus, total germ cell depletion in the testes of PND35 and adult NELF-B cKO mice reflects this penetrance and the requirement of NELF-B for spermatogonial survival/differentiation. We have now acknowledged this in the revised text.

5) The observation that paused Pol II is associated with sites of SPO11-mediated DSBs is potentially exciting. However, this association could be explained by independent correlation of SPO11 and paused PolII with open chromatin or other factors, rather than any mechanistic relationship between them. The correlation alone does not robustly support the conclusion that “loss of pausing in NELF-B cKO SG causes a collapse of promoter chromatin architecture, causing defects in SPO11-mediated DSB formation” (lines 455-457).

We agree that what we show in this work is a correlation and not causation or a clear mechanistic relationship between pausing and SPO11-mediated DSB formation. We have toned down the statement in question and modified other areas of the text to better reflect the strength of the data.

6) Methods relating to the 10x scRNA-seq data are relatively sparse. In addition, it is surprising that there are so few round spermatids in the control. This could be because of the juvenile stage of the testes or because of clustering artifacts. UMAP projections and/or violin plots of the markers used for cluster assignment should be shown in the supplement so the accuracy of cluster assignments can be more easily evaluated. The number of replicates should be specified and also how replicates (if any) were treated in downstream analysis.

We apologize for not including more information on the clustering approach. We have now added Supplemental Fig 5a (shown below for the reviewer), which shows the expression of the marker genes used for clustering across all cell types (modeled on a similar figure in a paper from the Hammoud lab), which we hope clarifies our approach.

Concerning the dearth of round spermatids, we agree with the reviewer that the age of the mice (PND24) played a role in the relative numbers of round spermatids being lower than spermatogonia or spermatocytes.

Figure S5a

Marker gene expression from scRNA-seq in control mice (n=4). Shown at left are the marker genes used to assign clusters to specific cell types. For this heatmap, signal for each gene was normalized within the row, to compare expression across cell types.

meiotic entry. However, we note that genes in cluster 5 experience maximal expression at a later stage, suggesting that a distinct pause release factor could be at play during later stages of spermatogenesis.

9) Some quantification should be provided for the Spo11 PRO-seq browser tracks shown in Figure 6i; as shown, it is hard to distinguish differences between the tracks from background noise.

We agree that the browser track shown was so zoomed in on the gene that it was difficult for a viewer to determine what level of signal is background noise. We have worked to make this clearer by zooming out more, so that the PRO-seq signal upstream of the Spo11 TSS is visible, with signal in this upstream region reflecting background PRO-seq signal. For quantification, each track is scaled such that 50 reads is the maximum in the track window, so that the height of the peaks near the Spo11 TSS, and within the gene body can be readily compared.

10) Figure panels 1k and S1d appear to be the same. If there is a difference, this should be better explained in the figure panel and legend.

We also note the number of replicates in main text (control n=4, TDP-43 cKO n=3, NELF-B cKO n=2), and added more detail to the methods section describing differential gene expression.

Minor comments:

7) “Mature spermatids” (line 24) is confusing; sperm are considered mature while spermatids are still maturing.

We agree. This has been changed.

8) It would be helpful for the authors to be more explicit about the assumptions regarding when in spermatogenic development pause release is thought to occur. Spermatocytes remain in meiotic prophase for about two weeks, and the STA-PUT sorted fraction includes cells from across this time frame. Transcriptional activation by pause release is thought to be important for rapid activation. In the scenario proposed by the authors, would pause release occur at meiotic entry as implied by its importance for SPO11 activation? If so, this could be more explicitly described in the discussion.

We have now clarified our model in the discussion. We envision that pause release occurs at cluster 4 genes at

Apologies for not making these clearer. Figure 1k shows the length of all RNAs in each cluster, whereas S1d focuses only on mRNAs. The supplementary figure addresses an important concern the reviewer raises above, about removing large gene classes that could be confounding our analyses of gene length.

11) In Figure 3, it is claimed that there is similar PRO-seq signal at promoter and gene body in clusters 1 and 2. This looks correct for cluster 1, but in cluster 2 there are quite a few genes with high signal at both gene body and promoter in SC and only at the promoter in SG.

The reviewer is correct. We have clarified in the revised text that these statements pertain primarily to cluster 1. As the reviewer astutely notes, there are differences between these two clusters, and it is cluster 1 that is more uniformly initiation limited. In agreement with this, in response to another reviewer's questions about the prevalence of the TATA motif, which is bound by the initiation factor TATA-binding protein (TBP), we find that clusters 1 and 6 are enriched in genes with TATA motifs.

12) In Figure 4b, there are some positively-staining cells toward the lumen in the cKO image at PND15, which is not a typical location for Sertoli cells. Is this nonspecific staining, or are these germ cells that escaped Cre excision?

As the reviewer correctly pointed out, these are germ cells that escaped Cre excision. This is now noted in the figure legend.

Reviewer #2 (Remarks to the Author):

This paper investigates the role of RNA polymerase II (Pol II) pausing during spermatogenesis and its impact on gene expression and meiotic processes. By combining bulk sequencing, PRO-seq, and single-cell RNA-seq techniques from wild-type (WT) mice and two conditional KO mouse strains: NELF or TDP, both known to affect Pol II pausing. The results highlight the essentiality of Pol II pausing in spermatogonia for the appropriate expression of genes crucial to spermatogenesis. Furthermore, the loss of NELF, an inducer of Pol II pausing, in immature germ cells leads to inhibited differentiation into round spermatids. Additionally, immunohistochemistry and scRNA-seq analysis in NELF and TDP43-deficient testes reveal an unexpected role of Pol II pausing in the regulation of meiosis and meiotic gene expression, including the double-strand break protein SPO11. This intricate interplay between RNA Pol II pausing and double-strand break formation adds a novel dimension to our understanding of spermatogenesis regulation. While further investigation is required to fully elucidate these mechanisms, this study presents intriguing preliminary findings that contribute significantly to the germ cell research community.

COMMENTS:

1. The observation that spermatid genes do not accumulate paused Pol II, even in spermatogonia or spermatocytes, is intriguing. Further investigation is needed to determine the differences in RNA Pol II complex recruitment between round spermatid genes and earlier germ cell genes. It would be interesting to explore whether these differences are related to TBP-dependent or TBP-independent gene regulation.

We thank the reviewer for suggesting this interesting analysis. TBP and TBP-like factor (TLF), which work at the step of transcription initiation (rather than pausing) have been implicated in spermatogenesis, with TLF being required for round spermatid function.

We find that genes in clusters 1 and 6 are particularly enriched in the TATA motif, suggesting that TBP/TLF play key roles in spermatogonia (when cluster 1 genes are highly active), and in round spermatids (when cluster 6 genes dominate). We note that although TLF has been reported not to tightly bind the TATA motif,

that the DNA structure, particularly a DNA bend elicited by the TATA motif facilitate interactions of TFIID with the promoter, also supporting initiation. Thus, the presence of TATA in cluster 1 and 6 genes fits very well with our results implying that these genes are more likely regulated at the step of initiation, rather than pausing. These results are now shown in Supplementary Figure 3a.

2. It would be beneficial to investigate the expression patterns of NELFb and TDP43 proteins/RNA to gain a better understanding of their roles in spermatogenesis. Examining their expression profiles in different germ cell populations and at various stages of development could provide valuable insights into their functions.

Figure S5b

Gene expression of NELF-B (left) or TDP-43 (right) in control mice (n=4) for the indicated cell type. Shown are cumulative counts from scRNA-seq for each cell type.

We agree. We have now investigated the expression of NELF-B and TDP-43 across all cell types in the scRNA-seq, in control animals. These results, shown at left and in new Supplementary Fig. 5b, support both proteins playing an important role early in male germ cell development. In particular, we see highest NELF-B expression in SG and SC where pausing is established and pause release is occurring. In contrast, we observe very low NELF-B expression in RS, in which we find that transcription initiation is the primary rate limiting step. This is consistent with our RNA-seq analyses, in which NELF-B is a cluster 2 gene, with gradually decreasing activity across differentiation.

In contrast, TDP-43 levels peak in SC and is a cluster 3 gene in our analyses. This is consistent with cluster 3 being enriched in genes involved in mRNA processing and post-transcriptional gene regulation (see Supplemental Figure S1b). We thank the reviewer for suggesting this interesting analysis.

3. In the absence of NELF or TDP43 mutants, it would be intriguing to determine if the paused spermatogonia (SPG) genes identified by bulk PRO-seq in wild-type samples are prematurely expressed or if the loss of pause leads to gene expression loss.

Additionally, it would be interesting to investigate whether the loss of NELF or TDP43 results in the accumulation of Pol II at a second pause site, as described in a previous study titled "NELF regulates a promoter-proximal step distinct from RNA Pol II pause-release." This finding would be particularly relevant in the context of double-strand break (DSB) formation.

It was a great suggestion for us to look at the expression of all cluster 4 genes (paused in SG, but not substantially expressed until SC) in the NELF and TDP-43 cKO animals. This analysis, shown below and in Fig 6g, demonstrates that as the reviewer suggests, the loss of pausing at these genes in NELF-B cKO animals leads to reduced gene activation in SC. Moreover, this revealed that this group of genes shows

lower expression in TDP-43 cKO SC than in SG. This striking result suggests that loss of TDP-43 might additionally destabilize these RNAs, causing increased RNA turnover.

We would love to be able to map the exact location of pausing in SG or SC depleted of NELF but are unable to obtain enough cells of the NELF-B cKO genotype to perform these assays. In order to perform PRO-seq we used 2×10^7 wild type SG, SC, and RS cells. SG were derived from 25 PND6-8 mice while SC and RS were

derived from 10 adult (3 month) mice. To do this in NELF cKO mice would be an enormous challenge. At best, there is a 1 in 8 frequency of finding an F^{-/-}, iCre (NELF cKO) male pup per litter, making it not feasible for us to meet the above cell number requirement of age-matched NELF-B cKO pups.

4. The proportion of spermatocyte cells in NELF and TDP43 mutants appear to differ. It would be informative to examine whether these knockout (KO) strains exhibit different stages of arrest. Providing the fraction of Leptotene, Zygotene, and Pachytene spermatocytes would enhance the understanding of the specific developmental stages affected by the NELF and TDP43 mutations. Synchronization of spermatogenesis in mutants could potentially increase the number of stage specific germ cells numbers available for PRO-seq analysis.

Figure S5d

Numbers of each germ cell type (including stages of SC) per genotype. Cell counts were normalized by the number of mice per genotype (Control n=4, NELF-B cKO n=2, TDP-43 cKO n=3).

This was also a great idea. We have now analyzed the PND24 testis scRNA-seq data of both NELF-B and TDP-43 cKO and noted that the impact on prophase I of meiosis occurs at different stages (Supplementary Fig. 5d). Loss of NELF-B caused a reduction in the zygotene stage (Z), and the trend continued in pachytene and diplotene spermatocytes. In contrast, a drastic reduction of cells occurred at the pachytene (P) stage in TDP-43 cKO mice, which is consistent with maximum expression of TDP-43 protein in pachytene spermatocytes (Reference #6).

5. Investigating whether the decrease in Spo11 levels in the NELF and TDP43 KO animals leads to changes in the number of DSBs during meiotic prophase in these mutants is crucial. As changes in SPO11 expression have been shown to modulate DSB numbers. Examining this relationship in the NELF and TDP43 mutants would provide valuable insights into the impact of Pol II pausing on DSB formation.

5. Investigating whether the decrease in Spo11 levels in the NELF and TDP43 KO animals leads

We agree and have done several additional experiments and analyses to address this question. First, we have looked specifically at genes involved in meiosis in the scRNA-seq data. We noted that Cluster 4 genes are enriched in Gene Ontology categories related to meiosis. These categories had inadvertently been collapsed under a category related to microtubules in the original manuscript but have now been separated back out. Strikingly, we find 91 Cluster 4 genes in the category Meiotic Nuclear Division, which are also well represented in our scRNA-seq (Supplementary Fig 6e. and below). Analysis of the expression patterns of these genes across cell types and genotypes in our scRNA-seq data revealed that these genes are: i) highly upregulated in control SC as compared to SG, as expected, ii) significantly less activated in NELF-B SC;

e Meiotic Nuclear Division

iii) not activated at all in TDP-43 SC. These findings highlight that Spo11 is not the only meiotic regulator that is disrupted in NELF-B or TDP-43 cKO animals. Indeed, among this list of 91 genes with perturbed activation are those proven to be critical for DSB formation and repair (for e.g., Mre11a, Spo11, Rad51c, Spata22, Meiob, Hormad1, and Brca2). Further, this includes genes involved in synaptonemal complex formation (Sycp1, Sycp2, Sycp3) and the formation of cross-overs (Msh4, Msh5, Rnf12, Mlh1, Mlh3, and Cntd1). These intriguing findings lead to the expectation that NELF-B mutant spermatocytes will

show defects in DSB repair and synaptonemal complex formation.

To test this interesting hypothesis, we performed immunofluorescence on meiotic chromosome spreads using antibodies to Sycp3 and gamma H2AX (Fig 8a). In WT pachytene spermatocytes the gamma H2AX signal was restricted to the sex body whereas in NELF-B mutant pachytene-like spermatocytes large areas of the nucleus retained gamma H2AX signal. Since gamma H2AX is recruited to DNA breaks, its persistence on autosomes suggested unrepaired DNA DSBs.

We then used an additional marker, RPA, to investigate this further (Fig 8b). RPA binds to the ssDNA ends caused by the DSBs. Notably, the number of RPA foci should decrease in pachytene spermatocytes as other proteins replace RPA. In contrast, in NELF-B cKO mice there were many more RPA foci persisting in the mutant pachytene-like spermatocyte compared to the WT control, indicating impaired DNA DSB repair. Imaging and quantification of this data are shown in Fig 8, with quantification also shown for the reviewer at right.

Fig. 8 NELF-B cKO mice have impaired DSB repair.

At Left: Quantification of relative γ H2AX area (γ H2AX/DAPI) between the control (n=40) and NELF-B cKO (n=52) from three biological replicates. P-value from t-test.

At right: Quantification of the RPA foci numbers at pachytene from control (n=51) and NELF-B cKO (n=46) from three biological replicates. P-value from t-test.

The larger area of γ H2AX foci and higher number of RPA foci in NELF-B cKO implies impaired DSB repair.

6. The correlation between SPO11-oligos and

PRO-seq enrichment in promoter proximal regions in Spermatogonia is intriguing. Exploring the instructive nature of these PRO-seq peaks and investigating whether sites of DSB changes occur in the NELF and TDP43 KO strains where pausing is lost would be valuable. Additionally, determining where these germ cells arrest in meiotic prophase and assessing whether these mutants phenocopy the SPO11 knockout would further enhance our understanding of the relationship between Pol II pausing, DSBs, and meiotic progression.

This is another great question, but unfortunately, we are not technically able to map the sites of DSBs in NELF-B cKO animals. However, we have addressed where these germ cells arrest in meiotic prophase. Analysis of scRNA-seq data showed that loss of NELF-B leads to a depletion in zygotene spermatocytes thus indicating the requirement of NELF-B early in prophase I.

7. Generating PRO-seq libraries from both wild-type and NELF mutants (and TDP43 mutants as well, although the number of Leptotene cells may be lower) would significantly strengthen the most exciting aspects of the paper. Synchronizing spermatogenesis stages could potentially help enrich the Leptotene/Zygotene stage, facilitating more comprehensive PRO-seq analysis and improving the robustness of the findings.

As explained above, it is not feasible for us to perform PRO-seq on NELF-B mutant germ cells. The reviewer is right in pointing out that synchronization is a great way to enrich a given population of germ cells. However, the number of germ cells required for PRO-seq (2×10^7) means that we need a large number of cKO pups. Treatment of a batch of PND2 pups with the RA-inhibitor to synchronize spermatogenesis is doable with WT but not with mutant mice. At best we obtain one male F^{-/-}, iCre NELF-B mutant pup per litter whose genotype will not be known until PND8. Thus, one must blindly treat many litters on PND2

with the RA-inhibitor in order to obtain synchronized NELF-mutant testes. Given that our protocol requires 2×10^7 cells per PRO-seq experiment, synchronization is not a feasible experiment.

Reviewer #3 (Remarks to the Author):

Kaye et al., in this manuscript, described the roles of RNA polymerase II (Pol II) pausing in spermatogenesis and regulation of meiosis. They found thousands of differentially expressed genes during spermatogenesis. Interestingly, various clusters of genes are differentially expressed in Spermatogonia (SG), Spermatocytes (SC), and spermatids (RS). PRO-seq analyses confirmed the transcriptional regulation of these clusters of genes during spermatogenesis. Accumulation of PRO-seq signals at promoter-proximal regions representing Pol II pausing correlated with gene activity during spermatogenesis. Furthermore, PRO-seq data showed an increased transcriptional initiation or faster release of paused Pol II complexes associated with overexpressed genes.

NELF-B conditional knockout (KO) mice showed vacuole formation and disorganization of meiotic cells as early as postnatal day 15 (PND15). Furthermore, the loss of germ cells and more vacuoles appeared at PND24. A further increase in germ-cell loss was correlated with aging in NELF-B KO mice. Single-cell RNA-sequencing (scRNA-seq) data showed a defect in germ cell maturation in NELF-B KO mice. This phenotype was similar to TDP-43 germ cell KO. Various genes were downregulated, including genes related to spermatid development and motility pathways, confirming the germ cell developmental defects in both NELFB and TDP-43 KO mice.

They further correlated the Pol II pausing (promoter activity) with sites of double-strand breaks. They found that NELF-B KO in SG both perturbs Spo11 gene activation as well as impairs the proper progression of meiosis.

This is a well-written manuscript. I have the following comments that could improve this manuscript further:

1) It's unclear if the gene size difference (Fig 1k) is statistically significant.

We apologize for not making this clear. We now show p-values on the graphs of gene size in Fig 1k and Supplementary Fig. 1d. Cluster 6 genes are significantly shorter than genes in all other clusters.

2) It's good to see RNA-seq and PRO-seq results corroborate; distinct mechanisms may be involved in gene regulation. It would be great to show a faster pause release of some representative genes (or genome-wide, if possible) using the Pol II decay curve after Triptolide treatments (time course).

We agree that these would be very cool experiments to perform, but unfortunately, they aren't feasible in this system given the number of cells we can realistically obtain. Reliable rate measurements from Triptolide time courses require multiple replicates and a series of time points, which are beyond the scope of what is doable in this study.

3) How Pol II pausing is impacted after NELF KO in germ cells? Looking at the cell numbers in Fig 5c, it could be possible to do PRO-seq or CUT&RUN using SG and SC cells (cells could be pooled) after NELF KO and comparing them with the control samples. Results from these experiments could further strengthen the notion that, indeed, Pol II is essential during spermatogenesis for proper gene expression and completion of meiosis.

We agree that these would be very cool experiments to perform, but unfortunately, they aren't feasible in this system. As noted above, there is a 1 in 8 frequency of finding an F^{-/-}, iCre (NELF cKO) male pup per litter, making it not feasible for us to meet the above cell number requirement of age-matched NELF-B cKO pups.

4) Data presented under the heading "Promoter activity in SG correlates with sites of double-strand breaks in SC" in the Result section is fascinating. Authors need to validate these findings by using their NELF KO models. Does disrupting Pol II pausing changes the sites of double-strand breaks?

It would be very insightful to address whether NELF-B loss has changed the sites of DSBs. However, only a few labs have been able to perform the Spo11 oligo sequencing experiment. Per Dr. Keeney's publication (Reference #40) describing the method, they have used 200 WT mice to perform the IP of Spo11 and subsequent oligo sequencing. As we hope the reviewer will understand, it would be near impossible to obtain 200 NELF-B mutant mice.

REVIEWERS' COMMENTS

Reviewer #1 (Remarks to the Author):

This revised manuscript has addressed my major concerns and is substantially improved. In particular, I appreciate the clearer and more explicit discussion of how the TDP-43 and NELF-B KO phenotypes differ, and justification for using both mutants. The study is rigorously done and adds useful and valuable new insight about the role of Pol II pausing in spermatogenesis and DSB formation. I just have a few minor comments that should be addressed in a final version of the manuscript:

1. I appreciate the detailed analysis and response to my question regarding normalization of RNA-seq data to 4C DNA content in spermatocytes. The authors' argument that all four copies may not be transcriptionally active in spermatocytes, and decision to maintain the original normalization strategy based on the analysis shown in the rebuttal, is fair. I suggest that the higher DNA content in spermatocytes should be noted in the Results section when the normalization strategy is discussed, and the rationale for normalization should be briefly explained there.
2. Line 36: It is not technically true that there are "no" treatment options for male infertility; multiple assisted reproduction options are available (IUI, IVF, ICSI, TESE) but these depend on the specifics of the infertility phenotype. It would be more accurate to say there are "limited treatment options available".
3. Line 43: The phrasing that there are three germ cell types is somewhat misleading; as the authors know, there are multiple subdivisions of the spermatogonia, spermatocyte, and spermatid stages that could each be considered cell types. A better phrasing might be "three major developmental stages".
4. In discussing the finding that genes expressed later in spermatogenesis tend to be shorter than those expressed earlier, it would be better to say "we speculate" rather than "we hypothesize" (line 145).

Reviewer #2 (Remarks to the Author):

The authors have addressed all concerns. The revised manuscript is really exciting and thought provoking. It's ready for publication.

Reviewer #3 (Remarks to the Author):

The authors have made significant improvements to the manuscript, addressing most concerns and providing justifications where additional data could not be included.

Response to critique:

We thank all three reviewers for their careful reading of our revised manuscript and the thoughtful and constructive suggestions made to improve this study. We have addressed all comments of Reviewer 1 and have edited the text of our manuscript. Reviewers 2 and 3 did not request changes.

Our point-by-point responses to Reviewer 1 comments are below. We hope that the reviewer agrees that our manuscript is stronger as a result of these changes, and we appreciate their input.

Reviewer #1 (Remarks to the Author):

This revised manuscript has addressed my major concerns and is substantially improved. In particular, I appreciate the clearer and more explicit discussion of how the TDP-43 and NELF-B KO phenotypes differ, and justification for using both mutants. The study is rigorously done and adds useful and valuable new insight about the role of Pol II pausing in spermatogenesis and DSB formation. I just have a few minor comments that should be addressed in a final version of the manuscript:

1. I appreciate the detailed analysis and response to my question regarding normalization of RNA-seq data to 4C DNA content in spermatocytes. The authors' argument that all four copies may not be transcriptionally active in spermatocytes, and decision to maintain the original normalization strategy based on the analysis shown in the rebuttal, is fair. I suggest that the higher DNA content in spermatocytes should be noted in the Results section when the normalization strategy is discussed, and the rationale for normalization should be briefly explained there.

We are glad that the reviewer agrees with our rationale for normalization. Per suggestion, we have now added in the Results section, “Our normalization strategy further included a correction for the haploid status of RS cells as compared to diploid SG and SC stages (see Methods). Although SCs have 4C DNA content following replication in meiosis, we could not find evidence that all genome copies were competent for transcription at this stage, so no additional corrections were made.” on line 91-95 in the revised manuscript.

2. Line 36: It is not technically true that there are “no” treatment options for male infertility; multiple assisted reproduction options are available (IUI, IVF, ICSI, TESE) but these depend on the specifics of the infertility phenotype. It would be more accurate to say there are “limited treatment options available”.

We agree with the reviewer. We have deleted “No” and changed the sentence to “limited treatment options available” on line 40 in the revised manuscript.

3. Line 43: The phrasing that there are three germ cell types is somewhat misleading; as the authors know, there are multiple subdivisions of the spermatogonia, spermatocyte, and spermatid stages that could each be considered cell types. A better phrasing might be “three major developmental stages”.

In agreement with the reviewer, we have changed the phrase to “major developmental stages” on line 47-48 in the revised manuscript.

4. In discussing the finding that genes expressed later in spermatogenesis tend to be shorter than those expressed earlier, it would be better to say “we speculate” rather than “we hypothesize” (line 145).

We agree and have changed the wording to “speculate” on line 156.

Reviewer #2 (Remarks to the Author):

The authors have addressed all concerns. The revised manuscript is really exciting and thought provoking. It's ready for publication.

We thank Reviewer 2 for their positive remarks.

Reviewer #3 (Remarks to the Author):

The authors have made significant improvements to the manuscript, addressing most concerns and providing justifications where additional data could not be included.

We thank Reviewer 3 for their positive remarks.

Overall, we are very happy that our manuscript reporting an important advancement in the field of transcriptional regulation of spermatogenesis was well-received by the Reviewers.